



# Combined sun-photometer/lidar inversion: lessons learned during the EARLINET/ACTRIS COVID-19 Campaign

Alexandra Tsekeri[1], Anna Gialitaki[1, 14, 20], Marco Di Paolantonio[2,21], Davide Dionisi[2], Gian Luigi Liberti[2], Alnilam Fernandes[3], Artur Szkop[3], Aleksander Pietruczuk[3], Daniel Pérez-Ramírez[4,5], Maria J. Granados Muñoz[4,5], Juan Luis Guerrero-Rascado[4,5], Lucas Alados-Arboledas[4,5], Diego Bermejo Pantaleón[4,5], Juan Antonio Bravo-Aranda[4,5], Anna Kampouri[1, 19], Eleni Marinou[1], Vassilis Amiridis[1], Michael Sicard[6,22], Adolfo Comerón[6], Constantino Muñoz-Porcar[6], Alejandro Rodríguez-Gómez[6], Salvatore Romano[7], Maria Rita Perrone[7], Xiaoxia Shang[8], Mika Komppula[8], Rodanthi-Elisavet Mamouri[9,10], Argyro Nisantzi[9,10], Diofantos Hadjimitsis[9,10], Francisco Navas-Guzmán[11,5], Alexander Haefele[11], Dominika Szczepanik[12], Artur Tomczak[12] Iwona S. Stachlewska[12], Livio Belegante[13], Doina Nicolae[13], Kalliopi Artemis Voudouri[1,14], Dimitris Balis[14], Athina A. Floutsi[15], Holger Baars[15], Linda Miladi[16], Nicolas Pascal[16], Oleg Dubovik[17], and Anton Lopatin[18]

[1]IAASARS, National Observatory of Athens, Athens, Greece
[2]Institute of Marine Sciences (ISMAR), Italian National Research Council (CNR), Rome - Tor Vergata, Italy
[3]Institute of Geophysics, Polish Academy of Sciences, Warsaw, Poland
[4]Applied Physics Department, University of Granada, Granada, Spain
[5]Andalusian Institute for Earth System Research (IISTA-CEAMA), Granada, Spain
[6]CommSensLab, Dept. of Signal Theory and Communications, UPC, Barcelona, Spain
[7]Mathematics and Physics Department, University of Salento, Lecce, Italy
[8]Finnish Meteorological Institute, Kuopio, Finland
[9]Cyprus University of Technology, Dep. of Civil Engineering and Geomatics, Limassol, Cyprus
[10]ERATOSTHENES Centre of Excellence, Limassol, Cyprus
[11]Federal Office of Meteorology and Climatology, MeteoSwiss, Payerne, Switzerland
[12]University of Warsaw, Faculty of Physics, Warsaw, Poland
[13]National Institute of R&D for Optoelectronics, Magurele, Romania
[14]Physics Department, AUTH, Thessaloniki, Greece
[15]Leibniz Institute for Tropospheric Research, Leipzig, Germany
[16]AERIS/ICARE Data and Services Center Villeneuve d'Ascq, France
[17]Laboratoire d'Optique Atmosphérique, Lille 1 Université de Lille - Science and Technology, Lille, France
[18]GRASP SAS, Villeneuve-d'Ascq, France
[19]Department of Meteorology and Climatology, School of Geology, Aristotle University of Thessaloniki, 54124 Thessaloniki, Greece
[20]Earth Observation Science Group, Department of Physics and Astronomy, University of Leicester, Leicester, UK
[21] Scuola di Ingegneria, Università degli Studi della Basilicata, Potenza, Italy
[22]Laboratoire de l'Atmosphère et des Cyclones, Université de la Réunion, Saint-Denis, France

*Correspondence to*: Alexandra Tsekeri (atsekeri@noa.gr)

**Abstract.** The European Aerosol Research Lidar Network (EARLINET), part of the Aerosols, Clouds and Trace gases Research Infrastructure (ACTRIS), organized an intensive observational campaign in May 2020, with the objective of monitoring the atmospheric state over Europe during the COVID-19 lockdown and relaxation period. Besides the standard operational processing of the lidar data in EARLINET, for seven EARLINET sites having co-located sun-photometric



observations in AERONET, a network exercise was held in order to derive profiles of the concentration and effective-column size distributions of the aerosols in the atmosphere, by applying the GRASP/GARRLiC inversion algorithm. The objective of this network exercise was to explore the possibility to identify the anthropogenic component and to monitor its spatial and

temporal characteristics in the COVID-19 lockdown and relaxation period. While the number of cases are far from being statistically significant so as to provide a conclusive description of the atmospheric aerosols over Europe during this period, this network exercise was fundamental to derive a common methodology for applying GRASP/GARRLiC on a network of instruments with different characteristics. The limits of the approach are discussed, in particular the missing information close to the ground in the lidar measurements due to the instrument geometry, and the sensitivity of the GRASP/GARRLiC retrieval

to the settings used, especially for cases with low AOD as the ones we show here. We found that this sensitivity is well-characterized in the GRASP/GARRLiC products, since it is included in their retrieval uncertainties.

## 1 Introduction

Preventive measures, mainly associated with public lockdowns and traffic restrictions, were imposed on a worldwide scale in an attempt to regulate the spread of the COVID-19 pandemic in early 2020. According to WMO's Air Quality and Climate

Bulletin (World Meteorological Organization, (2021), State of the global climate 2020, Geneva: WMO No-1264), the reduction in anthropogenic activities affected primary pollutant emissions, and hence air-quality on a global scale (Venter et al., 2020; Ghossiere et al., 2021). This unprecedented situation provided the opportunity to study the changes in atmospheric composition with respect to the anthropogenic aerosol component, which includes the aerosol particles originated from transportation, domestic heating, industrial facilities and long-range inter-continental transport, among others.

EARLINET (Pappalardo et al., 2014; Schneider et al., 2000) organized the "EARLINET/ACTRIS COVID-19 campaign" (https://www.earlinet.org/index.php?id=covid-19), with the aim to monitor the atmospheric state over Europe during the COVID-19 lockdown and relaxation period, in May 2020. EARLINET is part of ACTRIS (https://www.actris.eu), and is comprised by thirty-three permanent station-members that perform regular lidar measurements, following a predefined schedule for measurements, along with quality-assurance procedures, as these are established by the network. Moreover, the

network performs measurements during special atmospheric circumstances (e.g. dust transport; Pappalardo et al., 2009), and/or alerts on hazardous situations (e.g., volcanic eruptions; Pappalardo et al., 2013; Papagiannopoulos et al., 2020). During the one-month period of the EARLINET/ACTRIS COVID-19 campaign, the network demonstrated its capability to provide the vertical profiling of aerosols and clouds using multi-wavelength lidar measurements, in near-real time. In addition to the active remote sensing measurements, passive remote sensing measurements from sun-photometers were also acquired during the

campaign. The complementary use of active and passive remote sensing sensors has been proven to provide additional capabilities towards more accurate aerosol profiling and characterization (e.g., Lopatin et al., 2013).

Taking advantage of the data collected during the EARLINET/ACTRIS COVID-19 campaign, a subset of stations with collocated lidar and sun-photometers used the synergy of both types of observations in order to investigate the possible





reduction of the aerosols from anthropogenic activities above Europe, due to the COVID-19 lockdown. The characterization

of the anthropogenic aerosol component using remote sensing techniques is a challenging task. This is mainly due to the small load of anthropogenic particles in non-heavily polluted areas, and/or their mixture with aerosols originating from natural sources, such as windblown dust, marine aerosols or transported smoke. Lidars are the only instruments that can provide detailed vertically-resolved profiles of aerosol properties (e.g., Ansmann and Müller, 2005), but the detection suffers from observational constraints in the lowest part of the planetary boundary layer (PBL) (Kotthaus et al. 2023) where most of the

anthropogenic aerosols reside, due to instrument geometry (Chen et al., 2014; Navas-Guzmán et al., 2011; Wandinger and Ansmann, 2002).

Several lidar studies have highlighted the difficulty to distinguish between aerosols associated with anthropogenic activities and other species, mainly due to similarities in the optical properties measured by the lidars (e.g., Burton et al., 2012, 2013; Groß et al., 2013; Nicolae et al., 2018; Papagiannopoulos et al., 2018). For example, anthropogenic aerosols may be

misclassified as biomass burning due to their similar particle extinction-to-backscatter ratio (lidar ratio; $S$) and Ångström exponent (Å). In Müller et al. (2007) and in Janicka et al. (2017), the spectral dependence of $S$ between 355 and 532 nm was used to distinguish between biomass burning and anthropogenic particles, an option not available when using single-wavelength lidar systems or lidar systems that provide extinction measurements only at a single wavelength. The separation of anthropogenic aerosols and large dust particles can be less complicated when depolarization lidar measurements are

available, due to the strong depolarization signature of dust (Ansmann et al., 2012; Sugimoto and Lee, 2006; Tesche et al., 2009). This is though not the case for fine dust, which exhibits low depolarization values (Järvinen et al., 2016; Sakai et al., 2010; Szczepanik et al. 2021).

The synergy of lidar with sun-photometer measurements provide more advanced retrievals for profiles of particle microphysical properties, as shown for the LIdar-Radiometer Inversion Code (LIRIC) in Chaikovsky et al. (2018), and for the

Generalized Aerosol Retrieval from Radiometer and Lidar Combined data (GARRLiC), part of the Generalized Retrieval of Atmosphere and Surface Properties (GRASP), in Lopatin et al. (2013; 2021). Being applied to (most commonly) three-wavelength elastic backscatter lidar data, these algorithms can separate through fine and coarse species (LIRIC provides additional separation of spherical and non-spherical coarse particles, with a common refractive index for all species though). Tsekeri et al. (2017) have shown the limitations of these algorithms in complex atmospheric scenes containing anthropogenic

particles along with marine and dust particles. A common practice to derive a better characterization is to employ other types of observations (e.g., in-situ measurements) or atmospheric models to support the lidar, sun-photometer or combined lidar/sun-photometer analysis. For instance, in Deleva et al. (2019), forecast models and air-mass backward trajectories were used, complementary to lidar observations, to exclude the possibility of other aerosol types (i.e., dust or smoke) being mixed with a persistent layer of urban pollution found above Sofia, Bulgaria. Newly-developed aerosol classification algorithms take this

synergy a step further by making also use of land-surface coverage and additional satellite information of active fire regions, to avoid the inherited ambiguity in aerosol characterization based on optical properties alone (Mylonaki et al., 2021; Radenz et al., 2021).



Herein, we use a similar synergistic approach, in order to derive the microphysical properties of the anthropogenic aerosol component in different European areas. We use air-mass back-trajectory and emission sensitivity analysis, in order to verify
the absence of transported (fine) dust or smoke particles, thus the absence of mixtures with the fine particles of anthropogenic origin in the atmospheric scenes. We utilize the synergy of lidar and sun-photometer measurements in the advanced inversion scheme of GRASP/GARRLiC that provides columnar and vertically-resolved optical and microphysical properties of fine and coarse particles separately. In the absence of natural aerosols in the atmospheric scenes, the retrieved fine particles contain only anthropogenic aerosols. Due to the low aerosol optical depth (AOD) during the campaign, the retrieval uncertainty was
found to be high in some cases, particularly for the complex refractive index (Lopatin et al., 2013). Hence, herein we mainly focus on the retrieval of the columnar size distribution and the concentration profile of the anthropogenic particles, excluding the retrieved complex refractive index from our results.

While the objective of the EARLINET/ACTRIS COVID-19 campaign was the monitoring of possible changes in the load and properties of aerosols from anthropogenic activities, the objective of this study is the derivation of a common methodology for
applying the GRASP/GARRLiC algorithm on a network of different systems (with different configurations), along with the identification of the issues that should be solved to optimize this process. Coordinated observational activities within the framework of EARLINET have been organized also in the past as part of near-real-time operability demonstrations (D'Amico et al., 2015; Sicard et al., 2015), or during extreme events, such as the Eyjafjallajökull volcano eruption in 2010 (Ansmann et al., 2012; Pappalardo et al., 2013; Sicard et al., 2012). However, these efforts provided only the optical properties of the
particles. The present study is -to the best of our knowledge- one of the very few coordinated efforts on the European network level to provide concentration profiles and column-integrated microphysical properties of aerosol particles in the atmosphere (another example is provided in the work of Granados-Muñoz et al. (2016)).

The study is organized as follows: in Sect. 2 we present a brief overview of the measurement strategy during the EARLINET/ACTRIS COVID-19 campaign, along with the geographical distribution and the characteristics of the
participating EARLINET stations. In Sect. 3 we present the lidar and sun-photometer measurements used to derive the microphysical properties of the anthropogenic particles, along with the atmospheric transport models used to verify the absence of natural aerosols, and the GRASP/GARRLiC retrieval algorithm. Section 4 presents the methodology for applying the GRASP/GARRLiC algorithm on a network level. The retrieved microphysical properties of the anthropogenic particles are presented in Sect. 5. Section 6 presents the issues identified from running GRASP/GARRLiC on a network level, and Sect. 7
summarizes this work and presents the conclusions.

## 2 The EARLINET/ACTRIS COVID-19 campaign

The intensive observational EARLINET/ACTRIS COVID-19 campaign was the result of the EARLINET initiative to detect possible changes in the atmospheric composition and atmospheric aerosol load, during the COVID-19 pandemic outbreak in Europe. Twenty-one EARLINET stations participated (Fig. 1), performing measurements continuously or twice per day (at 12



and 20 UTC for two-hour intervals). All stations operated either automatic and/or remotely-controlled instruments. The lidar measurements were processed and Level 1 and 2 data were provided in near-real-time (ACTRIS ARES Data Centre, 2020). The Level 1 data include the attenuated backscatter (i.e., the calibrated range-corrected lidar signal) and the Level 2 data include the particle backscatter and extinction coefficients. Potential critical issues of the lidar systems, as well as quality assurance of the signals were assessed prior to the campaign, using standard quality-assurance procedures applied within EARLINET (Freudenthaler et al., 2018; Belegante et al., 2018). Sun-photometer measurements and related products for aerosol optical and microphysical columnar properties were provided by the Aerosol Robotic Network (AERONET) global sun-photometric network (Holben et al., 1998).

In the present analysis we used measurements from six out of the twenty-one EARLINET stations participating in the campaign (pink symbols in Fig. 1), and Payerne station (EARLINET station not participating in the campaign) based on:

- The collocation criteria of the sun-photometer and lidar measurements (i.e., a maximum of 30 min time difference and 1 km spatial difference between the sun-photometer and lidar measurements).

- The full overlap height of the lidar measurements (different for each system, as shown in Table 3, with maximum value of 1 km), which marks the lowest height of trustworthy profiles, since below this height the lidar backscattered signal is only partially collected from the receiving telescope (e.g. Wandinger and Ansmann, 2002).

The list with the characteristics of the stations participating with data in the current study is given in Table 1, including the EARLINET and AERONET station names, the latitude and longitude, the altitude, and the type of the surrounding area (i.e., rural, suburban, urban, background and industrial).

Regarding the selected cases, we first selected all cloud-free cases for which the sun-photometer and lidar measurements had +/- 30 minute difference, and we filtered out the cases with presence of fine natural aerosol particles in the atmosphere (i.e., smoke and dust particles). In order to identify the latter we used the modeling tools described in Sec. 3.3, along with lidar measurements of the volume and particle linear depolarization ratio (VLDR and PLDR, respectively), indicating non-spherical particles (i.e., dust particles), as discussed in Sect. 3.1. As shown in Table 2, the selected cases that contain mostly anthropogenic particles are 40% of the available cloud-free cases for the EARLINET stations used in the analysis.

## 3 Data and tools

### 3.1 Lidar data

EARLINET lidars used in this study employ Nd:YAG lasers that emit radiation pulses at high repetition rate (usually 20 Hz), recording the light backscattered from the constituents of the atmosphere (i.e., molecules, aerosols, clouds). The outgoing laser radiation is linearly-polarized and emitted simultaneously at multiple wavelengths (i.e., 355, 532, 1064 nm) as shown in Table 3. Moreover, many of those systems also employ Raman extinction channels at 387 and 607 nm to measure light backscattered from $N_2$ molecules, and provide profiles of the particle extinction coefficient at 355 and 532 nm, respectively (Ansmann et al., 1990). Finally, the cross-polarized backscattered light (with respect to the emitted light polarization) is typically recorded





at 355 and/or 532 nm, to derive profiles of the VLDR and PLDR. These profiles are used for identifying non-spherical particles in the atmospheric column (e.g., dust particles), which depolarize the backscattered laser light. Table 3 also reports the operation mode of the lidars during the campaign (i.e., continuous or measuring twice per day), and the system full overlap
height.

An example of the lidar measurements is shown for the continuous measurements provided by the Polly$^{XT}$-NOA lidar located at Antikythera EARLINET station, during the period of the campaign. Specifically, the attenuated backscatter coefficient at 532 nm is shown in Fig. 2a, and the VLDR at 532 nm is shown in Fig. 2b. The VLDR profile was used in our analysis along with atmospheric transport model simulations, to identify dust layers in the column (Fig. 2b).


In this study, we utilize two types of EARLINET lidar products:

    a) Level 1 data of the pre-processed, range-corrected signal at 355, 532 and 1064 nm, used as inputs in the GRASP/GARRLiC algorithm (Sect. 4.1).

    b) Level 2 data of the particle backscatter and extinction coefficients at 355, 532 and 1064 nm, used for the comparison
185         with the corresponding products from GRASP/GARRLiC algorithm (Sect. 4.2).

The majority of the Level 1 and 2 lidar products are processed with the Single Calculus Chain (SCC) algorithm (D'Amico et al., 2016; Mattis et al., 2016). The SCC is an automatic-analysis tool for lidar data, developed within EARLINET, that ensures the homogeneity and quality of adverse lidar system data, and facilitates the processing of lidar measurements in a fully automatic way. Only the observations from the Payerne station are processed with an in-house developed algorithm, since the
signals at 1064 nm were acquired with a ceilometer. To ensure sufficient noise suppression in the lidar signals, a time interval of two-hours and a vertical smoothing window of 60 m are selected for temporal and spatial averaging, respectively. Data with low signal-to-noise ratio (i.e., relative error of the lidar signals more than >50%), are not used in the current analysis.

The Level 1 data are derived by the SCC pre-processing module (ELPP module) that corrects the measured lidar signals for dead-time effects, trigger-delay, background subtraction and overlap, as described in detail in D'Amico et al. (2015; 2016).
The ELPP output files also contain the statistical uncertainty of the Level 1 data, along with the molecular atmosphere profiles, calculated using either radio-sounding or atmospheric numerical weather prediction models (e.g. from the Weather Research and Forecasting (WRF) Model; Marinou et al., 2019).

The Level 2 data are provided by the SCC ELDA module (Mattis et al., 2016), and include profiles of the particle backscatter and extinction coefficient (the latter available mainly for night-time measurements), along with the VLDR and PLDR (used as
an indication of the presence of non-spherical particles in this analysis). The optical property profiles are typically calculated either with the Klett-Fernald method (Fernald, 1984; Klett, 1981), or with the iterative method (Di Girolamo et al., 1995), and for some lidars with the Raman method (Ansmann et al., 1990). The latter method is more frequently applied during night-time, in the absence of strong background radiation. ELDA module also provides the statistical uncertainty of Level 2 data.



### 3.2 Sun-photometer data

The EARLINET stations that participated with measurements in this study were equipped with CIMEL sun-photometers, part of the AERONET global sun-photometer network. In absence of clouds and during daytime, CIMEL sun-photometers perform direct-sun irradiance measurements every 15 min, and almucantar measurements of sky radiance every hour. Regular calibration procedures are employed within the AERONET network to assure high-accuracy retrievals, with residuals of 0.01 and 5% for direct-sun and sky-radiance measurements, respectively. Sun and sky radiances at 440, 675, 870 and 1020 nm are

used to derive the AERONET products of the size distribution, complex refractive index and sphericity fraction of the aerosol particles along the atmospheric column (Dubovik and King, 2000; Dubovik et al., 2006). Additionally, the phase function, asymmetry factor, single scattering albedo, and absorption aerosol optical depth are provided at the same wavelengths.

The sun-photometer data used in the current study are:

   a) Level 1.5 data of direct-sun and sky-radiance measurements at 440, 675, 870 and 1020 nm, as input in the
GRASP/GARRLiC retrieval (Sect. 4.1).

   b) Level 1.5 data of the particle volume size distribution, for evaluating the corresponding results from the GRASP/GARRLiC retrieval (Sect. 4.2).

### 3.3 Atmospheric models

In order to identify the aerosol sources during the EARLINET/ACTRIS COVID-19 campaign, we used atmospheric model
simulations that helped us to exclude fine natural aerosol particles (dust and smoke), and focus only on the anthropogenic aerosol component. This was done through the identification of the origin of the aerosol particles using the FLEXible PARTicle dispersion model (FLEXPART; Pisso et al., 2019), the BSC-Dream dust model (Nickovic et al., 2001), and the Navy Aerosol Analysis and Prediction System (NAAPS). FLEXPART is a Lagrangian transport and dispersion model, which we used to identify the sources of the observed atmospheric layers, as shown for example in Fig. 4. The BSC-Dream dust model predicts
the atmospheric life cycles of desert dust. The NAAPS model produces forecasts of three-dimensional aerosol concentrations of sulfate, dust, and smoke aerosols in the troposphere.

### 3.4 GRASP/GARRLiC retrieval algorithm

The GRASP/GARRLiC retrieval algorithm performs a statistically-optimized inversion for both lidar and sun-photometer measurements, based on a multi-term least squares method, taking also into account a-priori information of the aerosol
properties (Lopatin et al., 2013; 2021). An example of a-priori knowledge is the smoothing constraints imposed on the retrieved volume size distributions, or on the spectral variability of the retrieved refractive index. The input of GRASP/GARRLiC consists of the pre-processed, range-corrected, normalized lidar signals (see Appendix A), along with the total optical depth (TOD) and the sky radiances at 440, 675, 870 and 1020 nm acquired from the sun-photometer. The lidar signals may be at one





or more wavelengths, where in this study we utilized the measurements at 355, 532 and 1064 nm, wherever they were available

(see Table 3).

Amongst others, the retrieved parameters include the columnar aerosol volume size distribution, the spectral complex refractive index at 440, 670, 870 and 1020 nm, and the profiles of aerosol concentration at 60 altitude levels, for both fine and coarse aerosol modes (see more details in Lopatin et al. (2013; 2021)). For each parameter the retrieval uncertainty is provided (Herrera et al., 2022) or can be calculated from GRASP/GARRLiC outputs (see Appendix B).

A limitation of the algorithm is the assumption of a fixed atmospheric profile (future versions of the algorithm are planned to properly address this issue). The fixed atmospheric profile may not adequately provide the backscatter and extinction profiles of the molecules in the atmosphere (i.e., similar issues are identified in Mona et al., 2009). Moreover, the gas absorption is also neglected within the channels. In this study only the elastic backscattering of the lidar signal is modelled (GRASP algorithm provides the capability to utilize Raman and depolarization lidar signals as well).

**4 Methodology**

**4.1 Methodology for applying GRASP/GARRLiC on a network level**

The methodology for applying the GRASP/GARRLiC algorithm on a network level is based on a two-step approach, first optimizing the parameters used to run the GRASP/GARRLiC retrieval (Section 4.1.1), and then evaluating the robustness of the results, through evaluating the sensitivity of the retrieval to the initial guess (Section 4.1.2).

**4.1.1 First step: optimizing the setting parameters of GRASP/GARRLiC retrieval**

In the first step of our methodology we define the appropriate combination of the setting parameters for running the GRASP/GARRLiC retrieval. More specifically, we optimize the a-priori smoothing constraints used in order to limit unrealistic oscillations (spikes) in the retrieved parameters. The smoothing is defined by (a) the order of finite difference of the function used to characterize the degree of non-linearity of (i) the volume size distribution, (ii) the spectral dependance of

the refractive index, and (iii) the height dependance of the concentration profile, for fine and coarse particles, and (b) by the corresponding Lagrange multipliers that determine the contributions of the smoothing into the solution (Dubovik and King, 2000; Dubovik 2011).

The definition of the optimum combination of the setting parameters is not a trivial task, and it may vary, e.g., for different atmospheric states. In the framework of this study we tried to derive a "global" optimum combination of the setting parameters

that could be used for all cases, tailored to each EARLINET station. This analysis was not conclusive, in part due to the low AOD of the case studies available in the current analysis (with AOD values lower than 0.15 at 500 nm for fine particles). Thus, herein, we derive the optimum combination of the setting parameters for each case and for each EARLINET station separately. The derivation is done by running multiple retrievals using different combinations for the values of the a-priori smoothing constraints. Table 4 shows the combinations of the values used for each a-priori smoothing constraint, for the case studies from



the different stations. Table S1 in the Supplement shows the optimum combination of the setting parameters selected for each case study. We should note that although we are interested in the retrieval of the fine mode, we need to optimize the smoothing constraints of the coarse mode as well, since they affect the retrieval of the fine mode properties.

The "optimum combination" for the setting parameters is the one providing the "optimum retrieval", considering the fitting of the measurements and the realistic values and oscillations of the retrieved parameters: The multiple GRASP/GARRLiC runs

that are executed for each case using different combinations of the a-priori smoothing constraints, provide an ensemble of solutions, from which we first select the "acceptable solutions". These are the solutions that have residuals within the uncertainty of the observations, i.e., 0.01 for TOD, 5% for sky radiances, and 30-50% (depending on the station) for the lidar/ceilometer signals. (We should note that the uncertainty of the lidar observations is considered here to be constant with height, although in reality, the uncertainty is height-dependent). The "optimum retrieval" is then selected from the ensemble

of the acceptable solutions, based on the requirements to have the least possible smoothing imposed and present no un-physical oscillations in any of the retrieved parameters. The latter criterion cannot be strictly quantified and is evaluated in a purely qualitative way, thus is mostly subjective, based on scientific experience.

To demonstrate our methodology, we show here an example of a case study from Antikythera station on May 25, 2020. Figure 3a shows the time-height evolution of the total attenuated backscatter coefficient at 532 nm, from 12:00 to 17:00 UTC, which

depicts that the majority of the aerosol load is found (from the ground) up to almost 4 km. The low PLDR values, ranging between 0.02 and 0.04 at 532 nm (not shown here), reveal that the particles are close to spherical, thus no dust particles are present. The FLEXPART source-receptor analysis in Fig. 4 shows that the air-masses that arrive in Antikythera on this day, follow North-Eastern and North-Western routes and for a total of 5 days preceding the observations, they mostly reside above urban areas and the sea. Consequently, it is expected that the atmospheric column consists of a mixture of marine and

anthropogenic particles. The marine particles are considered to be mainly coarse particles, thus are expected to have a minimum effect on the load of the fine mode, which consists mainly of anthropogenic particles. According to the collocated sun-photometer measurements, the AOD at 500 nm does not exceed the value of 0.15 and the fine aerosol fraction is higher than 80% (Fig. 3b). All the above advocate the presence of (mostly) fine mode particles, which are considered here to be the anthropogenic component we aim to retrieve.

The lidar measurements used as input for the GRASP/GARRLiC retrieval, are time-averages of the elastic lidar signals at 355 and 532 nm between 13:15 and 15:00 UTC (indicated with the red rectangular in Fig. 3a), achieving in this way sufficient noise suppression. The sky radiances and the TOD measurements of the sun-photometer are taken at 14:25 UTC (purple line on Fig. 3a), in order to overlap with the lidar observations.

We used the lidar and sun-photometer measurements from this case study and we applied the first step of our methodology:

we run ~2500 retrievals of GRASP/GARRLiC, using ~2500 combinations of a-priori smoothing constraints, with values shown for the case study "AKY" in Table 4. The retrievals that fitted the measurements within the uncertainty of the observations (i.e., 0.01 for TOD, 5% for sky radiances, and 30% for the lidar signals) are considered the acceptable solutions for this case. Figure 5 shows an example of the fitting of the lidar and sun-photometer observations for one of the acceptable solutions,



shown in Fig. 6. Finally, the optimum retrieval is selected amongst the acceptable solutions, based on the requirement to have
the least smoothing imposed and the absence of un-physical oscillations in the retrieved parameters. The latter is evaluated in
a purely qualitative way, as discussed above. The optimum retrieval for the volume size distribution and the concentration
profile of the anthropogenic component at Antikythera EARLINET station on May 25, 2020, along with the corresponding
retrieval uncertainty, is shown in Fig. 6 and 7 (blue line). The values of the a-priori smoothing constraints used for the optimum
retrieval are provided in Table 5.

A notable feature in Fig. 6 is that all of the acceptable solutions are within the retrieval uncertainty of the optimal solution.
This indicates that the variability of the solutions due to the smoothing constraints used to run the GRASP/GARRLiC retrieval,
is already provided in the GRASP/GARRLiC product, as part of the retrieval uncertainty of the solution.

### 4.1.2 Second step: sensitivity of the GRASP/GARRLiC retrieval to initial guess

In the second step we examine the stability of the optimum solution with respect to the values adopted as initial guess for the
retrieval. Although this step does not constitute a thorough investigation of the stability of the retrieval, it provides a first
quality check. Specifically, we generate ~100 (random) initial guesses and we run the retrieval using the optimum combination
of the a-priori smoothing constraints defined in step 1. In order for the optimum solution to be characterized as stable, the
retrievals performed with the random initial guesses should fall within its retrieval uncertainty. Figure 8 shows the results of
step 2 for the optimum solution for the case study from Antikythera EARLINET station on May 25, 2020. All the retrievals
that are calculated with different first guesses, fall within the retrieval uncertainty of the optimum solution, and thus verify its
robustness.

An interesting feature in Fig. 8 is the absolute difference of each solution calculated with variable initial guess, with respect to
the optimum solution, for the concentration profile. As shown in Fig. 8d, the underestimation (overestimation) of the
concentration profile within the lidar overlap region, due to the missing lidar information at this region, results in the
overestimation (underestimation) of the concentration profile at higher heights. This highlights the fact that the algorithm tries
to compensate for the over/under-estimation of the concentration profile below the overlap region and preserve a constant
value for the total aerosol concentration. The latter is provided mainly from the sun-photometer measurements, which contain
the information of the total column.

### 4.2 Comparison of GRASP/GARRLiC results with SCC and AERONET products

A qualitative evaluation of the retrieved microphysical and optical properties of the anthropogenic particles is done through
their comparison with the corresponding products of the volume size distribution from AERONET and the extinction and
backscatter coefficient profiles derived from SCC with the Klett method.

An example of the comparison of the retrieved volume size distribution with the AERONET product is provided for the case
study from Antikythera station on May 25, 2020 in Fig. 9, showing a good agreement, within the retrieval uncertainty of the
GRASP/GARRLiC product. In principle, the information content included in the lidar measurements, relative to the size





distribution, is extremely low compared to the one contained in the sky radiances measured from the sky/sun-photometer. Therefore, we do not expect the addition of the lidar measurements in the retrieval to have a strong impact on the derived aerosol microphysical properties.

Since we consider only daytime measurements, the SCC particle extinction coefficient profiles are calculated using the particle

backscatter coefficient profiles derived from ELDA and a constant pre-defined lidar ratio ($S$) value, which is characteristic for each specific scene. The extinction coefficient uncertainty is calculated through error propagation that includes both the uncertainty of the backscatter coefficient (provided directly by ELDA), along with the assumed uncertainty of the pre-defined $S$ value. The GRASP/GARRLiC backscatter and extinction coefficients are calculated from the retrieved microphysical properties, as shown in **Error! Reference source not found.** and **Error! Reference source not found.**, using the retrieved

concentration profile ($VD$), AOD ($\tau$) and lidar ratio ($S^G$), for fine ($f$) (i.e. anthropogenic) and coarse ($c$) particles. Here the superscript G is used to denote GRASP/GARRLiC retrieved properties, z denotes to altitude and $\lambda$ to wavelength.

$$\beta_\lambda^G(z) = \frac{\tau_{\lambda,f}^G \cdot VD_f^G(z)}{S_{\lambda,f}^G} + \frac{\tau_{\lambda,c}^G \cdot VD_c^G(z)}{S_{\lambda,c}^G} \tag{1}$$

$$a_\lambda^G(z) = \tau_{\lambda,f}^G VD_f^G(z) + \tau_{\lambda,c}^G VD_c^G(z) \tag{2}$$

The uncertainty of the particle backscatter and extinction coefficients retrieved from GRASP/GARRLiC, is calculated as shown in Appendix A.

Figure 10 provides an example comparison between the extinction and backscatter coefficient profiles of GRASP/GARRLiC and SCC products, for the case study from Antikythera station on May 25, 2020. The comparison shows a good agreement within the retrieval uncertainties of the SCC and GRASP/GARRLiC products. Both the particle backscatter and extinction profiles at 355 nm exhibit larger differences compared to the respective profiles at 532 nm. This is expected since the molecular scattering contribution is higher in the shorter wavelengths and thus, the effect of an assumed constant atmospheric profile in

the GRASP/GARRLiC retrieval is more prominent in the UV.

## 5 Results

The GRASP/GARRLiC retrievals of the volume size distribution and concentration profile of the anthropogenic component above Europe during the relaxation period of the Covid-19 lockdown are provided in Fig. 11 and 12, respectively. For the size distribution, the evaluation with the corresponding AERONET product is also included in each plot. Due to the low number

of cases, the results are not statistically significant. The results for the concentration profiles (Fig. 12) present a wide range of concentrations, along with large retrieval uncertainties. We should also note that they depend on the different overlap of the different lidar systems.





Details about the atmospheric state for each case study used in our analysis based on the lidar range-corrected signal quicklooks and the transport model simulations, along with the evaluation of the retrieved particle backscatter and extinction coefficient

with the corresponding products SCC, are provided in the Supplement.

Figure 13c, 14 and 15 show histograms of the retrieved center-of-mass (COM) height of the layer, the volume concentration, and the effective radius of the anthropogenic component over Europe, during the COVID-19 lockdown and the relaxation period. The COM was calculated from the retrieved concentration profile using the following equation (Siomos et al., 2017):

$$COM = \frac{\int z \cdot c \cdot dz}{\int c \cdot dz} \tag{3}$$

Where $z$ and $c$ correspond to altitude and concentration respectively.


Figures 13a and b illustrate the effect on the COM calculation of the limited information below the full overlap region, for different overlap heights. In order to investigate this effect we construct three different concentration profiles for calculating COM (see example in Fig. 13b, for the case study from Antikythera EARLINET station on 25 May 2020): "$h_m$" (blue) is the concentration profile without accounting for its retrieval uncertainty, "$h_a$" (pink) is the concentration profile taking into account

the maximum value above the full overlap height and the minimum value below the full overlap height. "$h_b$" (dark blue) is the concentration profile taking into account the minimum value above the full overlap height and the maximum value below the full overlap height. These profiles are provided for each case in the Supplement, and the COM calculated for each profile is plotted against the overlap height in Fig. 13a. Figure 13a shows that for the different cases herein, and different overlap heights, "$h_m$" has values of 1.25-1.75 km with a median value of 1.5 km, "$h_a$" has a median value of 1.61 km, and "$h_b$" has a median

value of 1.27 km. We see that there is no clear dependence of the calculated COM on the overlap height, and that the mean value and standard deviation of the COM calculated from the "$h_m$" concentration profile (Fig. 13c), characterizes well the distribution of values derived for "$h_a$" and "$h_b$".

The volume concentration is ~0.01-0.03 $\mu m^3 \mu m^{-2}$, with a median value of ~0.015 $\mu m^3 \mu m^{-2}$ (Fig. 14a), and the retrieved values are mostly similar with the corresponding AERONET products (Fig. 14b).

The retrieved effective radius is ~0.1-0.15 $\mu m$ (Fig. 15a). Figure 15b shows that the retrieved values are lower than the AERONET products for most of the cases used in our analysis (i.e. we compare the retrieved effective radius with the effective radius of the fine mode "REff-F" in the "size distribution parameters" inversion product of AERONET). This may be attributed to higher information content, compared to the AERONET, for the fine particles included in the GRASP/GARRLiC retrieval, due to the presence of lidar measurements at 355 nm. Another possible reason is the way the molecular contribution to the

signal is represented in the forward model, since a constant density profile for gasses is adopted. Given the spectral dependence of the molecular scattering properties, the signal at 355nm should be the one that is more affected by this assumption, and consequently, the retrieval of the smaller particles for GRASP/GARRLiC. Thus, in terms of the size distribution, an incorrect estimation of the molecular scattering contribution is likely to affect the retrieved size distribution of fine particles.





## 6 Issues identified from running GRASP/GARRLiC on a network level

The use of the GRASP/GARRLiC retrieval to assess the possible changes in aerosol particles related to anthropogenic activities above Europe during the EARLINET/ACTRIS COVID-19 campaign is to our knowledge one of the very few network exercises dedicated to derive profiles of particle microphysics above Europe. During this study, we identified the case studies (Sect. 2), data, modeling tools providing auxiliary information (Sect. 3), as well as the methodological tools that are required to derive the microphysical properties of aerosols from the combination of lidar with sun-photometer measurements on a

network level (Sect. 4). This process also involved the identification of the issues that a network study like this entails. These issues are grouped by the topics regarding the selection of the study cases, the aspects of the inversion approach and the evaluation of the results, and they are discussed in the following subsections.

### 6.1 Selection of the study cases

The **spatial and temporal collocation criteria** applied for the selection of suitable case studies, are a maximum of 30 min

acquisition time difference and 1 km spatial distance between the sun-photometer and lidar measurements (Sect. 2). These fixed threshold selection criteria are based on empirical knowledge of the optimum time and spatial difference between the different measurement datasets (i.e., Papagiannopoulos et al., 2016), and need to be re-evaluated for each station in the network, in order to take into account the effective spatio-temporal variability of the aerosols properties. In principle, the spatio-temporal variability is expected to depend on the local geographical characteristics of the site, as well as on the atmospheric conditions.

Moreover, some case studies were not used in the analysis, due to the **high full-overlap height** (>1km) of the corresponding lidar measurements. The lack of information in the lowermost atmospheric layers is a critical issue, given the typical vertical distribution of aerosols, and especially for the anthropogenic aerosols that are investigated herein. By selecting though only the systems with full overlap below 1 km, we only reduced the sensitivity to the assumptions adopted to handle this issue. A discussion on the impact of the missed information in this range of the atmosphere is provided in Sect. 6.2.

### 410 6.2 Aspects of the inversion approach

The experience during this campaign highlighted a strong dependence of the solution from the **smoothing constraints used to perform the inversion**. In Sect. 4 we present a detailed analysis of our methodological approach towards defining the optimum smoothing constraints based on objective criteria, but we recognize that there are few aspects that could be addressed in addition. First, the definition could also be based on a synthetic dataset for which the solution is known. Moreover, there

should be a dedicated study on whether it is possible to have a unique setting of constraints for the whole network or to define the settings based on the characteristics of the measurements (e.g., aerosol type, load, vertical distribution, temporal variability).

Regarding the sensitivity of the retrieval on the **first guess used for the inversion**, it was found to be not that significant, compared to the corresponding sensitivity of the smoothing constraints. Although herein we have used the first guess provided



as the default option for GRASP/GARRLiC retrieval, it may be preferable to define a first guess closer to the solution based on climatological data and/or previous days solutions.

We should highlight again that although the GRASP/GARRLiC retrieval was found to be sensitive to both setting parameters (i.e., smoothing constraints and first guess), the resulting variability was well-described by the retrieval uncertainty of the solution, as provided by the algorithm. Thus, the uncertainty provided for GRASP/GARRLiC product also accounts for the

uncertainty due to the sensitivity of the algorithm to different setting parameters.

Moreover, the retrieval depends also on the input dataset, and issues may arise from the way the **low-quality data** are handled. Herein we removed the low-quality data (i.e., lidar signals with noise >50% or cases where the sky-radiances from the sun-photometer were recorded in a small number of viewing angles) from the input dataset (losing completely the information), and for all other data we accounted for their quality through the covariance matrix of the observations. In future studies we

plan to screen out low-quality observations that do not provide any information to the retrieval, but incorporate the information of the low-quality data that contain useful information for the retrieval (providing their quality through the covariance matrix). The definition of the corresponding quality thresholds will be derived through inversion of synthetic datasets of measurements. Regarding the **covariance matrix**, herein we assumed it to be diagonal, with a constant relative uncertainty for each type of observation (i.e., AOD, radiances and lidar measurements). GRASP/GARRLiC provides the capability to use a height-

dependent non-diagonal covariance matrix for the lidar signals (Herrera et al., 2022). In future studies we should also take into account realistic uncertainties for the lidar measurements, instead of a constant relative uncertainty, although these are not always easy to characterize.

Similar to the removal of the low-quality data, we have also removed all **measurements below the full overlap height**. In this scenario GRASP/GARRLiC uses an homogeneous layer of constant concentration to characterize the missing information of the vertical distribution from the lidar signals at these heights. In case the real vertical distribution is different, this may

introduce discrepancies with the photometer measurements, which are sensitive to the vertical distribution in the lowermost atmospheric level. In future studies we should try to estimate independently the overlap function (Sasano et al., 1979; Tomine et al., 1989; Dho et al., 1997; Wandinger and Ansmann, 2002; Guerrero-Rascado et al., 2010; Di Paolantonio et al., 2022) and to use as input in GRASP/GARRLiC the lidar signals that are corrected for the overlap effect. A different solution could be

also for GRASP/GARRLiC to allow different vertical distributions for the overlap heights, as for example an exponential distribution with a given scale height. This would give a range of solutions and somehow, considering also the uncertainty estimated by GRASP/GARRLiC, the sensitivity to the approach adopted to handle the missing lidar information. Finally, in principle, GRASP/GARRLiC may be upgraded to retrieve the overlap function, including it in the forward model and in the state vector. The information for the retrieval may be provided utilizing the temporal multipixel approach (Lopatin et al., 2021),

through providing a set of profiles with a variability of aerosol vertical distribution, assuming constant instrument setup (thus, constant overlap function).

Finally, a potential critical issue in the current forward model of GRASP/GARRLiC is the **molecular scattering** representation. The forward model assumes a standard atmospheric density model to estimate the molecular scattering



contribution to the measured signal at the different wavelengths. Given the range of wavelengths of the signals analyzed, the
assumption of a constant molecular scattering is likely to generate artifacts. At low AODs this is likely to provide artifacts as
compared to the use of more precise temperature and pressure profiles. Using a user-provided atmospheric profile is planned
to be included in the future in GRASP/GARRLiC.

**6.3 Evaluation of results**

Section 4 provides a description of the methodology followed for the evaluation of the results adopting two approaches:
-    Consistency of the simulated signals against the input ones (Sect. 4.1)
    -    Soundness of the retrieved geophysical results (Sect. 4.2)

When evaluating the results it is necessary to define a quantitative metric to interpret the relevance of obtained differences. In
absence of a known solution, the only way to obtain a quantitative metric is by comparing the simulated vs the input signals.
Although such convergence can be obtained also with solutions with low geophysical soundness, it constitutes the limit of the
available information in the measurements, in absence of auxiliary information.

Regarding the comparison of our results with the products from EARLINET SSC and AERONET, this may imply the
assumption that there is no added value in the synergetic use of observations. However, the EARLINET SCC products are
derived using the Klett-Fernald algorithm, which is an independent method as it does not depend on the forward model or the
numerical inversion approach utilized within GRASP/GARRLiC and thus can be considered as an evaluation for the latter.
For the columnar properties, although AERONET and GRASP algorithms share many similarities, the additional information
coming from the lidar signals is missing from the stand-alone sun-photometer inversions. In future studies we plan to augment
the datasets used for the evaluation of our results, using e.g. (ground-based and/or airborne) in-situ measurements.

**7 Summary and conclusions**

Within the framework of the EARLINET/ACTRIS COVID-19 campaign, a subset of simultaneous and co-located AERONET
data was selected to be processed with GRASP/GARRLiC algorithm in order to provide additional information with respect to
the standard EARLINET products, such as aerosol columnar size distribution and concentration profiles. In principle, such
additional information was planned to be used to identify anthropogenic aerosols and to monitor their spatial and temporal
variability during the COVID-19 lockdown and relaxation period of May 2020.

By applying selection criteria, first on the compliance of the station (i.e., collocation with AERONET measurements), and
successively on the quality of the measurements (i.e., overlap height criteria, cloud-free atmospheres), and minimizing the
contribution of natural aerosols in the fine mode (i.e, fine desert dust and smoke particles), there was a drastic reduction in the
data availability. Subsequently, the obtained results have a low representativity and cannot be used to depict properly the
spatio-temporal variability of aerosol composition during the COVID-19 lockdown and relaxation period over Europe.
Nevertheless, we proceeded with the network exercise to apply GRASP/GARRLiC to measurements performed by lidar





systems with different technical characteristics, located in different regions, with the aim to identify the fundamental and propaedeutic issues on the use of GRASP/GARRLiC on the network level. This exercise is particularly helpful for defining the limitations of applying GRASP/GARRLiC on a network level, since the low-load conditions during the COVID-19 lockdown and relaxation period impose a stringent test to the algorithm's performance.

First, in addition to the input measurements and associated uncertainties, GRASP/GARRLiC results depend also on the setting

parameters used to run the retrieval (e.g., smoothing constraints and first guess). Although the sensitivity of the results is higher to the smoothing parameters than to the first guess used, the resulting variability of the solution due to both is included in the GRASP/GARRLiC product, as part of the estimated retrieval uncertainty. This is true for the solutions that reproduce the measurements within the measurement uncertainty. Thus, finding the optimum setting parameters for GRASP/GARRLiC retrieval may be non-trivial, especially since they seem to depend on the different stations within the network and on different

atmospheric conditions. Even so, reproducing the measurements (although using different setting parameters) is sufficient for finding the solution, since in this case the variability due to the use of different setting parameters is included in the retrieval uncertainty of the solution.

Second, from the point of view of input observations, a main issue is the overlap function. This may be handled in principle, by modifying GRASP/GARRLiC to retrieve the overlap function from the synergy of lidar and photometer data. In the future,

the requirement of ACTRIS for overlap of <300m will reduce the impact of this issue. Moreover, we did not exploit the full information provided by the EARLINET lidars (i.e. Raman and depolarization measurements in Table 3), although GRASP provides this capability.

Third, concerning the GRASP/GARRLiC forward model, a potential source of uncertainty has been identified in the constant atmospheric density profile adopted. Next GRASP versions are planned to include the capability of changing the atmospheric

profile by the user.

Fourth, the impact of combining the sun-photometer with lidar measurements with GRASP/GARRLiC is evaluated by comparing the obtained GRASP/GARRLiC products against the ones derived from single instrument measurements, namely the AERONET and the EARLINET/SCC products. We expect that the lidar information does not significantly influence the retrieval of aerosol columnar microphysical properties. This is observed for several cases, however there are few cases where

a change in the size distribution, in particular of the smaller particles, is obtained combining the instruments with respect to the sun-photometer-only (AERONET) product. A possible explanation may be the additional spectral information introduced by the lidar measurements at 355 nm, or it may be a bias due to the assumption of constant atmospheric density profile.

It should be noted that the above conclusions are obtained considering the capability of reproducing correctly the input measurements. For an ill-posed problem optimizing the inversion based uniquely on the capability to reproduce the input

measurements should be further supported by evaluation of the results with independent datasets (e.g., in-situ data), and/or by performing numerical exercises based on synthetic measurements.

Recognizing the value of the homogenization of the GRASP/GARRLiC retrieval within the framework of ACTRIS, AERIS/ICARE has been developing a new tool for automatic GRASP/GARRLiC retrievals, utilizing combined measurements



of EARLINET lidars and AERONET sun-photometers. The issue of the "optimum setting parameters" for each station has not

been solved as of yet for the automated retrieval. Currently, the same parameters are used for all stations. Moreover, at this stage, the evaluation of the results is done through visual inspection for e.g. unphysical oscillations etc, whereas evaluation with SCC products may be included in the future.

Overall, this study highlights the potential of utilizing GRASP/GARRLiC on a network level, with the aim of the effective characterization of the atmospheric state with respect to the microphysical properties of aerosol particles and their

concentration profiles. It also identifies the associated issues that require a more thorough investigation of their impacts, or new developments to the algorithm, along with recognizing the need for acquiring independent datasets for the evaluation of the results or to perform ad-hoc studies with synthetic datasets.

**Appendix A**

**A1: Normalization of range-corrected lidar signals**

The actual number of points in the range-corrected lidar signals used herein, varies with the selection of maximum and minimum altitude as well as the vertical resolution used during signal pre-processing in the SCC. For the EARLINET COVID-19 campaign the pre-processing vertical resolution and maximum altitude of the profiles was selected to be at 60 m and 15 km respectively, for all stations, while minimum altitude varies depending on the full overlap height of each system. The latter was carefully defined prior to the campaign, based on quality assurance tests performed on each system following

Freudenthaler et al. (2018). This procedure led to range-corrected signals of approximately 350 – 500 altitude points.

This number had to be scaled down before the signals are used as input to the GRASP/GARRLiC retrieval, in order to:

    i)    Reduce the excessively high number of retrieved parameters, since the altitude points of the retrieved concentration profiles are the same with the altitude points of the range-corrected lidar signals.

    ii)   Reduce noise contamination of lidar signals in high altitudes.

iii)  Decrease calculation time.

The procedure followed is described in Sect. A1, as well as in Lopatin et al. (2013) as *"decimation of lidar signals in logarithmic scale over altitude"* and additional details can be found in GRASP-Open webpage (https://www.grasp-open.com).

**A2: Lidar signals decimation and normalization**

In order to move to a logarithmic altitude/range scale and reduce lidar signal and lidar altitude vectors to points in a

logarithmically equidistant manner, we first define each profile point ($h_i$) in a logarithmic scale ($Z_i^{lg}$):

$$Z_i^{lg} = \log(Z_i) \tag{A1}$$

The logarithmic scale step $\Delta_z$ can be calculated from min ($Z_{min}^{lg}$) and max ($Z_{max}^{lg}$) altitude as:



$$\Delta_z = \frac{\left(Z_{max}^{lg} - Z_{min}^{lg}\right)}{N_z} \tag{A2}$$

With $N_z$ being the total number of logarithmically equidistant points in the profile (herein $N_z = 60$).

Then, each logarithmic altitude point ($h_k$) can be calculated as:

$$h_k = Z_0^{lg} + (k-1)\Delta_z, k = 1 \dots N_z \tag{A3}$$

Finally, for each $h_k$ the lidar signal profile and lidar altitude vector are averaged as follows:

$$A_k = \frac{\sum_{j=1}^{n} A_j(h_k, h_{k+1})}{n} \tag{A4}$$

Where $n$ is the total number of points inside logarithmic altitude ranges and $A_k$ denotes to lidar signal profile.

In order to account for discrepancies between different instruments, the last step is to normalize the lidar signals between min and max altitude following the equation:

$$A_k' = \frac{A_k}{\int_{z_{min}}^{z_{max}} A_k dz} \tag{A5}$$

## Appendix B: Calculation of the GRASP/GARRLiC retrieval uncertainties for the retrieved aerosol backscatter and extinction coefficients

Equations B1 and B2 provide the formulas utilized for the calculation of GRASP/GARRLiC backscatter and extinction coefficient uncertainty.

$$ln\left(a_{\lambda,k}^G(z)\right) = ln(VD_k^G(z)) + ln\left(\tau_{\lambda,k}^G\right) \to \sigma_{ln\left(a_{\lambda,k}^G(z)\right)}^2 = \sigma_{ln\left(VD_k^G(z)\right)}^2 + \sigma_{ln\left(\tau_{\lambda,k}^G\right)}^2 \tag{B1}$$

$$ln\left(\beta_{\lambda,k}^G(z)\right) = ln\left(VD_k^G(z)\right) + ln\left(\tau_{\lambda,k}^G\right) - ln\left(S_{\lambda,k}^G(z)\right) \to \sigma_{ln\left(a_{\lambda,k}^G(z)\right)}^2 = \sigma_{ln\left(VD_k^G(z)\right)}^2 + \sigma_{ln\left(\tau_{\lambda,k}^G\right)}^2 + \sigma_{ln\left(S_{\lambda,k}^G(z)\right)}^2 \tag{B2}$$

Where $k$ denotes to fine, coarse mode.

On a first order approximation, the standard deviation of the logarithms can be treated as the relative error so that and we

finally obtain:

$$\sigma_{ln\left(i_\lambda^G(z)\right)} = \frac{\sqrt{\left(i_{\lambda,f}^G(z)\right)^2 \cdot \sigma_{ln\left(i_{\lambda,f}^G(z)\right)}^2 + \left(i_{\lambda,c}^G(z)\right)^2 \cdot \sigma_{ln\left(i_{\lambda,c}^G(z)\right)}^2}}{i_{\lambda,f}^G(z) + i_{\lambda,c}^G(z)} \tag{B3}$$

Where $i$ denotes to extinction ($a$) or backscatter ($\beta$) coefficient profile.

**Data availability**

All the available EARLINET lidar data during the COVID-19 pandemic can be found in the ICARE webpage
(https://www.icare.univ-lille.fr/) and are published under: https://doi.org/10.21336/gen.w3w1-j222. AERONET L1.5
retrievals used for results validation, are publicly available at the webpage: https://aeronet.gsfc.nasa.gov/

**Supplement link**

To be added.

**Author contribution**

AT and AG organized and coordinated the work presented herein and structured the manuscript. AT, AG, MDP, DD, GLL,
AF, AS, AP, DPR, MJGM, JLGR formulated the methodology for running GRASP/GARRLiC on a network level (with
guidance from AL) and performed the GRASP/GARRLiC retrievals presented herein. MDP, DD, GLL, ISS provided valuable
guidance on the structure and content of the manuscript. AK, AS and DPR performed the atmospheric models simulations.
AT, AG, AF, AS, AP, EM, VA, MS, AC, CMP, ARG, SR, MRP, XS, MK, RF, WK, DS, ISS provided the high-quality lidar
and sun-photometer measurements used herein. All authors contributed with comments on the manuscript and reviewed the
final version.

**Competing interests**

Some authors are members of the editorial board of AMT. The peer-review process was guided by an independent editor, and
the authors have also no other competing interests to declare.

**Special issue statement**

This manuscript is part of the special issue "Quantifying the impacts of stay-at-home policies on atmospheric composition and
properties of aerosol and clouds over the European regions using ACTRIS related observations
(ACP/AMT inter-journal SI)".

**Acknowledgements**

We are grateful to EARLINET (https://www.earlinet.org/, last access: 12 April 2023) and ACTRIS (https://www.actris.eu, last
access: 12 April 2023) for the data collection, calibration, processing and dissemination. The authors would like to
acknowledge the use of GRASP inversion algorithm software (http://www.grasp-open.com) in this work. We thank the



AERONET team and all PIs and Co-Is and their staff for establishing and maintaining the 21 AERONET sites used in this investigation and making the data available for the community. This research was supported by data and services obtained from the PANhellenic Geophysical Observatory of Antikythera (PANGEA) of the National Observatory of Athens (NOA). Limassol's station acknowledge the 'EXCELSIOR': ERATOSTHENES: EXcellence Research Centre for Earth Surveillance and Space-Based Monitoring of the Environment H2020 Widespread Teaming project (www.excelsior2020.eu). The 'EXCELSIOR' project has received funding from the European Union's Horizon 2020 research and innovation programme under Grant Agreement No 857510, from the Government of the Republic of Cyprus through the Directorate General for the European Programmes, Coordination and Development and the Cyprus University of Technology. Kuopio station acknowledges the support of the Academy of Finland (projects nos. 310312 and 329216). The Granada station acknowledges the Spanish national project PID2020-120015RB-100, PID2020-117825GB-C21, PID2020-117825GB-C22 and PID2021-128008OB-I00funded by MCIN/AEI/10.13039/501100011033/FEDER "A way of making Europe", and projects A-RNM-430-UGR20, PP2022.PP.34, P20-0136, P18-RT-3820, H2020-MSCA-RISE-2017 'GRASP-ACE', ACTRIS-IMP, ATMO-ACCESS (grant agreement No 101008004), AEROMOST (Junta de Andalucia, ProExcel_00204), the Scientific Unit of Excellence: Earth System (UCE-PP2017-02) and the national infrastructure programs EQC2019-006192-P and EQC2019-006423-P.

## Financial support

This research was financially supported by D-TECT (Grant Agreement 725698) funded by the European Research Council (ERC) under the European Union's Horizon 2020 research and innovation program, and by PANGEA4CalVal (Grant Agreement 101079201) funded by the European Union       . Barcelona site acknowledges funding from the REALISTIC project (Grant Agreement No 101086690) funded by the European Union's Horizon Widera 2022 Talents program. Warsaw site acknowledges the financial support from the European Commission H2020 grant ACTRIS IMP (no. 871115).

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

**Table 1.** EARLINET and AERONET stations used for the characterization of the anthropogenic aerosol component in Europe, during the EARLINET COVID-19 campaign: name, location (Lat/Lon), altitude above sea level (a.s.l.), and type of surrounding area.

| EARLINET station ID and location | AERONET site | Lat (ºN), Lon (ºE) | Altitude (m.a.s.l.) | Area type |
|---|---|---|---|---|
| AKY - Antikythera, Greece | Antikythera_NOA | 35.86, 23.31 | 193.0 | Rural (maritime) |
| BRC - Barcelona, Spain | Barcelona | 41.39, 2.12 | 125.0 | Urban |
| COG - Belsk, Poland | Belsk | 51.83, 20.78 | 190.0 | Rural |
| KUO - Kuopio, Finland | Kuopio | 62.73, 27.55 | 190.0 | Rural |
| SAL - Lecce, Italy | Lecce_University | 40.33, 18.10 | 30.0 | Suburban |
| WAW - Warsaw, Poland | Warsaw_UW | 52.21, 20.98 | 112.0 | Urban |
| PAY - Payerne, Switzerland | Payerne | 46.82, 6.93 | 491.0 | Rural |

**Table 2.** The total number of measurements during the EARLINET/ACTRIS COVID-19 campaign and the selected cases presented in this
study: Total number of cases with Level 2.0 (quality assured) lidar backscatter profiles in EARLINET database. Total number of sun-photometer Level 1.5 inversions from AERONET database. Cloud-free lidar/sunphotometer collocated cases. Selected cases, containing mostly anthropogenic particles.





| EARLINET station ID | Total number of Level 2.0 lidar backscatter profiles (daytime and nighttime) | Total number of sun-photometer Level 1.5 inversions | Cloud-free collocated cases | Selected cases, containing mostly anthropogenic particles (based on data and back-trajectory analysis) |
|---|---|---|---|---|
| AKY | 70 | 97 | 9 | 2 |
| BRC | 428 | 74 | 6 | 4 |
| COG | 8 | 46 | 2 | 1 |
| KUO | 184 | 106 | 2 | 2 |
| SAL | 75 | 39 | 3 | 2 |
| WAW | 117 | 20 | 4 | 2 |
| PAY | 17 (*) | 81 | 12 | 3 |

(*Only Level 1.0 profiles)


**Table 3.** Characteristics of the lidar and sun-photometer measurements used for the characterization of the anthropogenic aerosol component in Europe, during the EARLINET/ACTRIS COVID-19 campaign: EARLINET station ID, lidar name, operation mode, wavelength of measured backscattered light, wavelength of Raman measurements (for extinction products), depolarization measurements, full overlap height above ground level (a.g.l.), wavelengths of sun-photometer measurements. The specific wavelengths used for the GRASP/GARRLiC
analysis for each station are highlighted in bold.

| EARLINET station ID | | AKY | BRC | COG | KUO | SAL | WAW | PAY |
|---|---|---|---|---|---|---|---|---|
| **Lidar name** | | Polly$^{XT}$-NOA | UPCLidar new+depol | Belsk LiDAR | Polly$^{XT}$-FMI | UniLE LIDAR | Polly$^{XT}$-WAW | Customized system |
| **Continuous operation** | | X | | | X | X | X | X |
| **Elastic backscatter wavelength (nm)** | **355** | **X** | **X** | **X** | **X** | **X** | **X** | **X** |
| | **532** | **X** | **X** | **X** | **X** | **X** | **X** | |
| | **1064** | X* | **X** | **X** | **X** | **X** | **X** | **X** |
| **Inelastic backscatter (Raman) wavelength (nm)** | **355** | X | X | | X | X | X | |
| | **532** | X | X | | X | X | X | |
| **Depolarization wavelength (nm)** | **355** | X | X | | X | X | X | |
| | **532** | X | X | | X | X | X | |
| **Full overlap height** | | 800 | 600 | 500 | 900 | 800 | 600 | 1000 |





| (m a.g.l.) | | | | | | | |
|---|---|---|---|---|---|---|---|
| **Sun-photometer wavelengths (nm)** | 340, 380, **440**, 500, **675**, **870**, **1020**, 1640 | 340, 380, **440**, 500, **675**, **870**, **1020**, 1640 | 340, 380, **440**, 500, **675**, **870**, **1020** | 340, 380, **440**, 500, **675**, **870**, **1020**, 1640 | 340, 380, **440**, 500, **675**, **870**, **1020**, 1640 | 340, 380, **440**, 500, **675**, **870**, **1020** | 340, 380, **440**, 500, **675**, **870**, **1020**, 1640 |

\* not operational during the campaign

**Table 4.** List of the a-priori smoothing constraints, and the corresponding range of values used for finding the optimum combination for the cases studied herein, for each EARLINET station. The selected optimum combination of the smoothing constraints is provided in Table S1 in the Supplement. OFD is the Order of Finite Difference, LM is the Lagrange multiplier, FM is the fine mode and CM is the coarse mode.

| | Smoothness constraints | | | | | | | | | | | |
|---|---|---|---|---|---|---|---|---|---|---|---|---|
| **EARLINET station ID** | **Volume size distribution** | | | **Real part of refractive index** | | | **Imaginary part of refractive index** | | | **Concentration profile** | | |
| | OFD | LM FM | LM CM | OFD | LM FM | LM CM | OFD | LM FM | LM CM | OFD | LM FM | LM CM |
| **AKY** | 3 | 0.005 0.05 0.5 | 0.05 0.1 0.5 | 1 | 30 50 100 | 30 50 100 | 1 | 0.1 0.3 0.5 | 0.1 0.3 0.5 | 3 | 0.005 0.05 | 0.005 0.05 |
| **BAR** | 3 | 0.005 0.05 0.5 | 0.005 0.05 0.5 | 1 | 10 20 80 | 10 20 80 | 1 | 0.5 1.0 1.5 | 0.5 1.0 1.5 | 3 | 0.001 0.05 | 0.001 0.05 |
| **COG** | 3 | 0.01 1 2 | 0.01 0.5 2 | 1 | 1000 | 1000 | 1 | 10 | 10 | 3 | 0.00001 | 0.00001 |
| **KUO** | 3 | 1 10 100 | 0.005 0.01 0.1 | 1 | 200 400 500 | 200 400 500 | 1 | 0.5 1.0 1.5 | 0.5 1.0 1.5 | 3 | 0.05 0.5 | 0.05 0.5 |
| **SAL** | 3 | 0.01 0.1 0.3 | 0.01 0.1 0.3 | 1 | 100 300 | 100 300 | 1 | 2 10 | 2 10 | 3 | 0.00001 0.1 10 | 0.00001 0.1 10 |
| **WAW** | 3 | 0.1 1.0 10 | 0.5 1.0 10 | 1 | 1000 1500 2000 | 1000 1500 2000 | 1 | 0.5 1.0 2.0 | 1.0 2.0 2.5 | 3 | 0.05 0.5 | 0.05 0.5 |
| **PAY** | 3 | 0.01 0.3 1 2 | 0.01 0.1 0.3 0.5 | 1 | 300 1000 | 300 1000 | 1 | 2 10 | 2 10 | 3 | 0.00001 0.001 0.5 | 0.00001 0.001 0.5 |





**Table 5.** The a-priori smoothing constraints used for the retrieval of the volume size distribution, the spectral refractive index and the concentration profile of the anthropogenic component at Antikythera station on May 25, 2020. OFD is the order of finite difference, LM is the lagrange multiplier, fm is the fine mode and cm is the coarse mode.

| Volume size distribution | | | Real part of refractive index | | | Imaginary part of refractive index | | | Concentration profile | | |
|---|---|---|---|---|---|---|---|---|---|---|---|
| OFD | $LM_{fm}$ | $LM_{cm}$ | OFD | $LM_{fm}$ | $LM_{cm}$ | OFD | $LM_{fm}$ | $LM_{cm}$ | OFD | $LM_{fm}$ | $LM_{cm}$ |
| 3 | 0.005 | 0.05 | 1 | 30 | 30 | 1 | 0.1 | 0.1 | 3 | 0.05 | 0.05 |

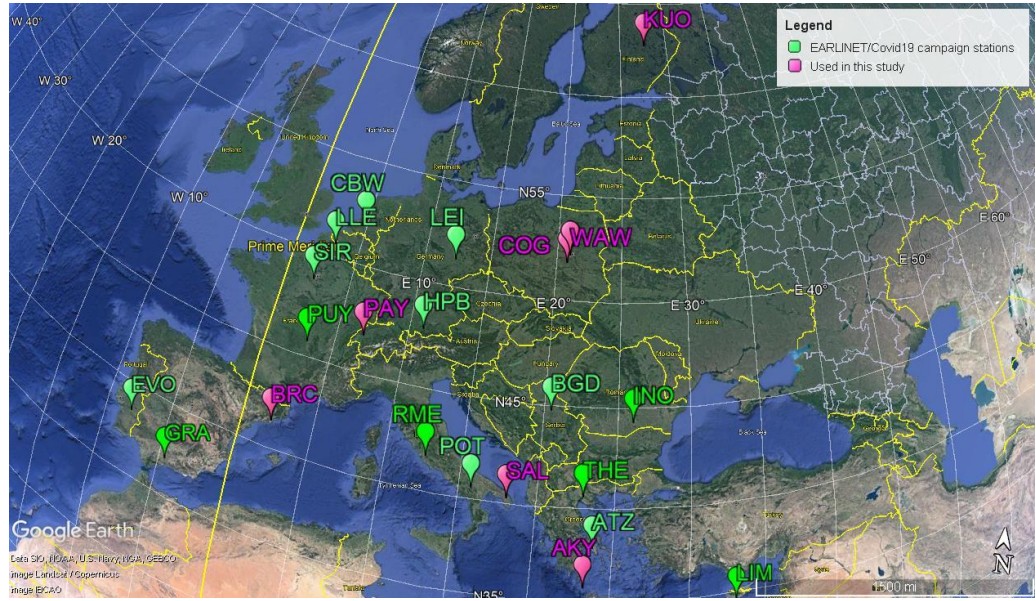


**Figure 1: The 21 EARLINET stations that participated in the COVID-19 campaign. The stations that were used for the characterization of the anthropogenic aerosol component in Europe are shown in pink and the rest of the stations in green. Payerne station data were also used in the present analysis, though the station did not participate in the campaign. The 3-digit code name of each station can be found in the EARLINET webpage: https://www.earlinet.org/index.php?id=105. Map created using Google Earth.**




**Figure 2: Time-height plots of continuous lidar measurements from the PollyXT-NOA lidar at the Antikythera (AKY) EARLINET station in Greece, for the whole period of the EARLINET/ACTRIS COVID-19 campaign (May 2020). a) Attenuated backscatter coefficient at 532 nm. b) VLDR at 532 nm. Altitude is provided in km a.s.l..**




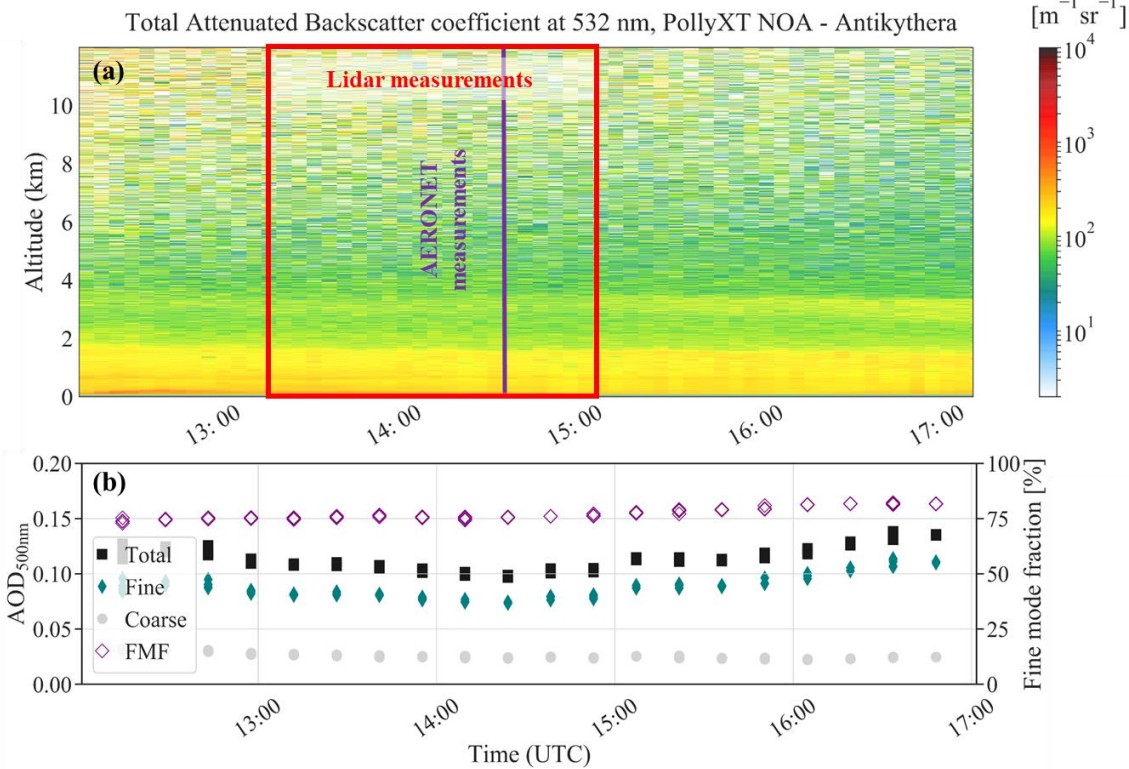

Figure 3: a) The total attenuated backscatter coefficient at 532 nm, for the case study at Antikythera EARLINET station, on May 25, 2020, at 12:00-17:00 UTC. Altitude is provided in km a.s.l. The time window of the lidar measurements used for the GRASP/GARRLiC retrieval presented herein is marked with the red rectangular. The time of the collocated measurements from the sun-photometer used in the retrieval is shown with a purple line. b) The collocated AOD at 500 nm for all particles (total; black squares), fine particles (green diamond) and coarse particles (gray circles), provided by AERONET. The fine mode fraction is also shown (purple diamonds).





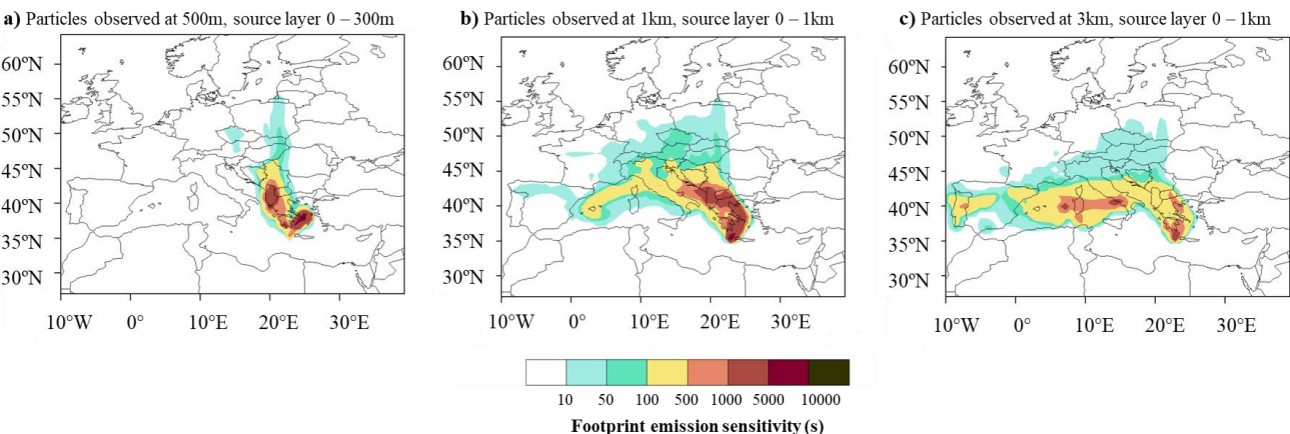

**Figure 4: 5-days backward FLEXPART-WRF calculation of footprint emission sensitivity in [sec], for the particles that arrive at Antikythera EARLINET station, on May 25, 2020 (14:25 UTC) at receptor heights a) 500m, b) 1km and c) 3km. The footprint emission sensitivity represents the residence time – the total time an air parcel spent over a certain surface and below the reception height, and is a first hint of the type of the aerosol load of the air parcel. The longer an air parcel resides close to the surface, the more likely it will acquire the aerosol footprint of the surface.**

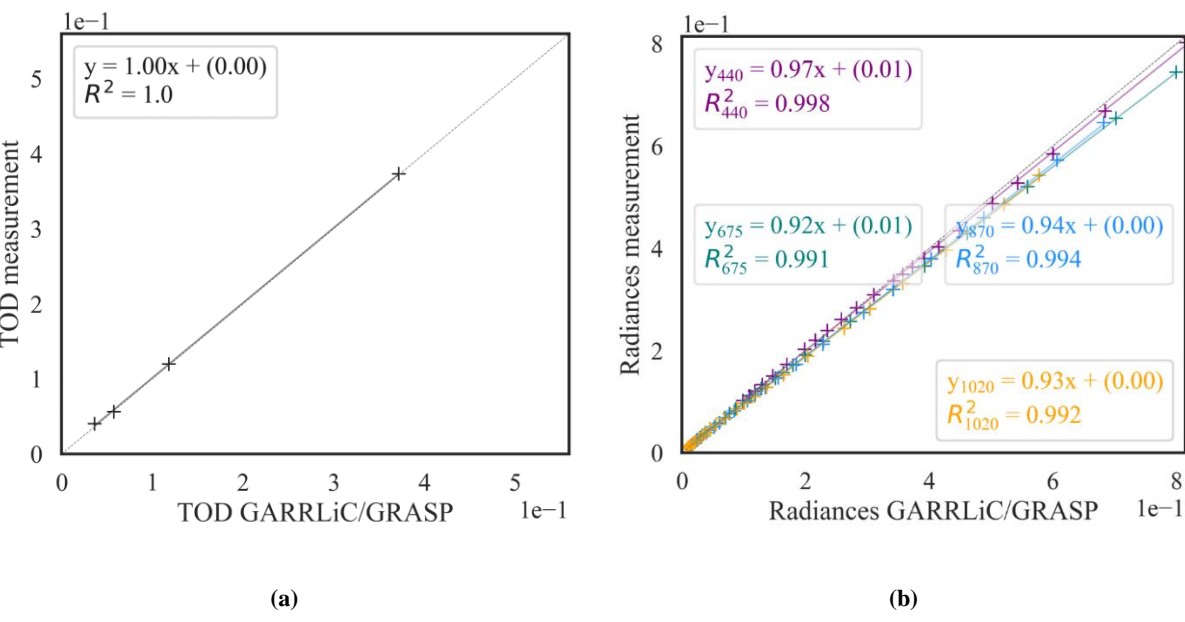

(a)  (b)



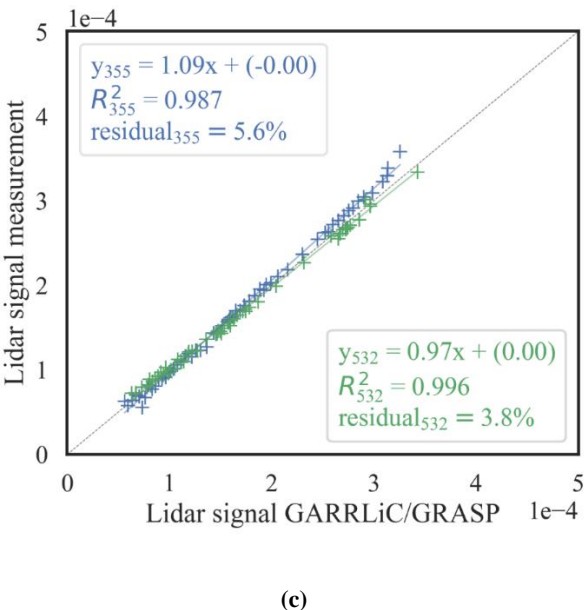

**(c)**

**Figure 5: Example of the GRASP/GARRLiC fitting of the lidar and sun-photometer observations, for the case study from Antikythera station on May 25, 2020: a) the TOD at 440, 675, 870 and 1020 nm, b) the sky radiances at 440, 675, 870 and 1020nm, and c) the normalized, range-corrected signals at 355 and 532 nm.**


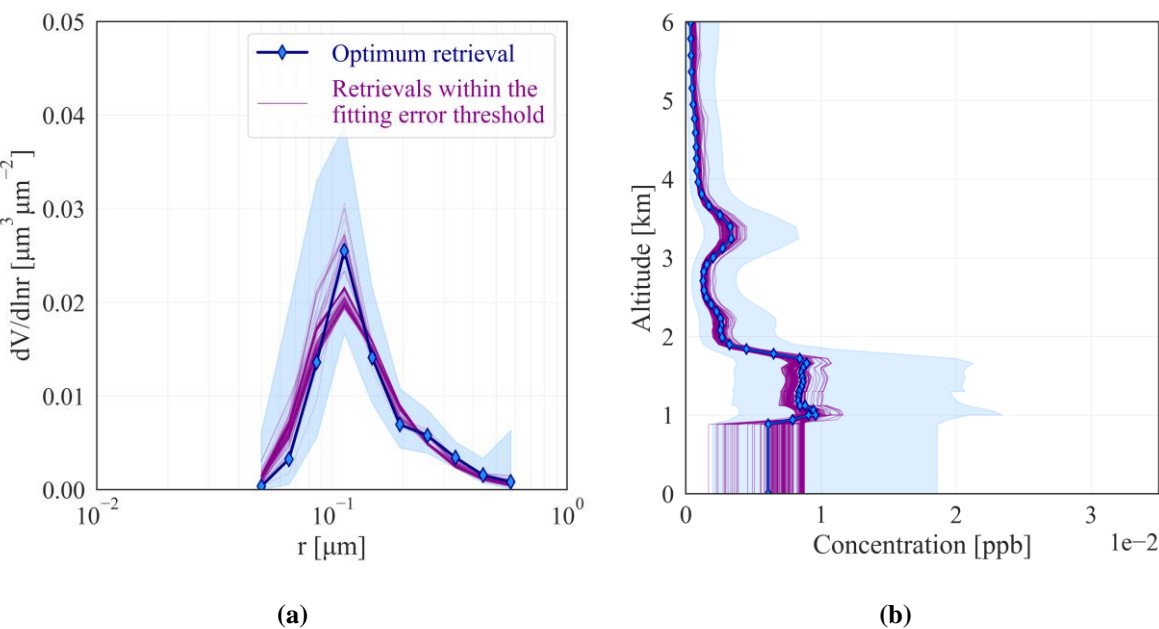

**(a)** **(b)**



**Figure 6: The a) volume size distribution and b) volume concentration profile of the anthropogenic component for the case study from Antikythera station on May 25, 2020 (altitude in km a.s.l.). For both plots, purple lines represent all the solutions derived with variable a-priori smoothing constraints for which the retrieval residuals fall within the uncertainty of the measurements, thus they are acceptable solutions. The blue line represents the optimum solution, provided with its retrieval uncertainty (blue shaded area).**


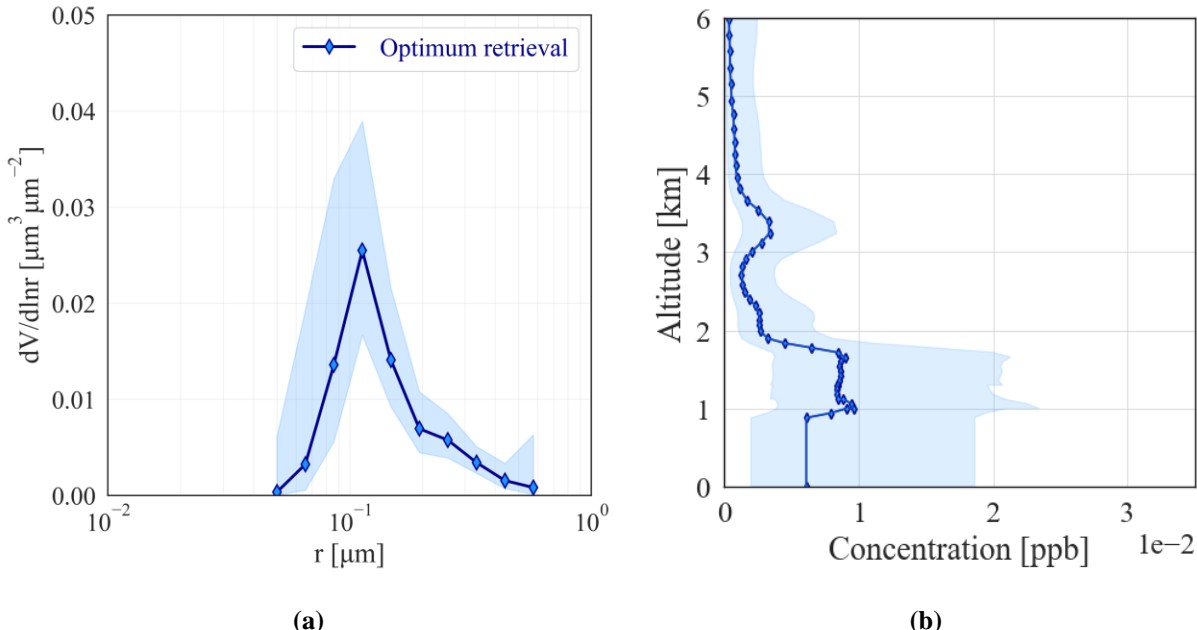

(a)        (b)

**Figure 7: The optimum solution, selected among the acceptable solutions shown in Fig. 6, for the case study from Antikythera EARLINET station on May 25, 2020: a) The size distribution (blue line) of the anthropogenic component, with the respective retrieval uncertainty (blue shade). b) The concentration profile of the anthropogenic component (blue line), with the respective retrieval uncertainty (blue shade) (altitude in km a.s.l.).**






(a)

(b)

(c)

(d)


**Figure 8: Test for the robustness of the optimum solution in Fig. 7, for the case study from Antikythera EARLINET station on May 25, 2020: a) the size distribution and b) the concentration profile of the anthropogenic component. The optimum solution for a) and b) is shown with blue, whereas the solutions calculated using different initial guesses are shown with purple. The retrieval uncertainty of the optimum solution is shown with blue shade. The absolute difference of each solution calculated with variable initial guess,**



with respect to the optimum solution, is shown in c) for the size distribution and d) for the concentration profile. Altitude in
concentration profiles is provided in km a.s.l.

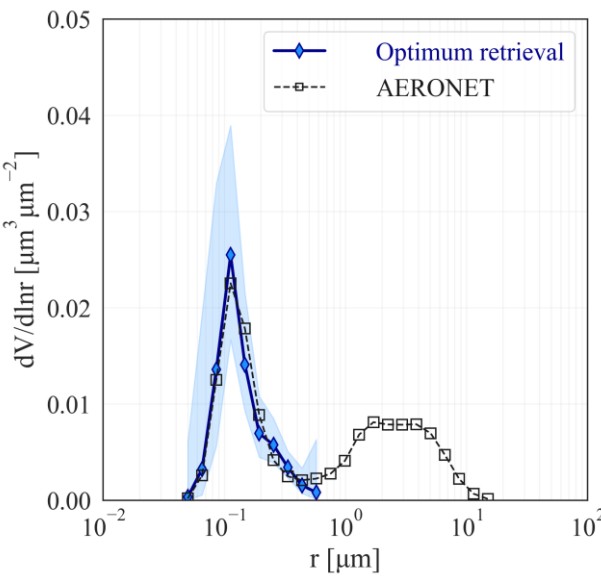

**Figure 9: Evaluation of the retrieved volume size distribution (blue line) of the anthropogenic component, with the AERONET
volume size distribution (black line with squares), for the case study from Antikythera EARLINET station on May 25, 2020.**

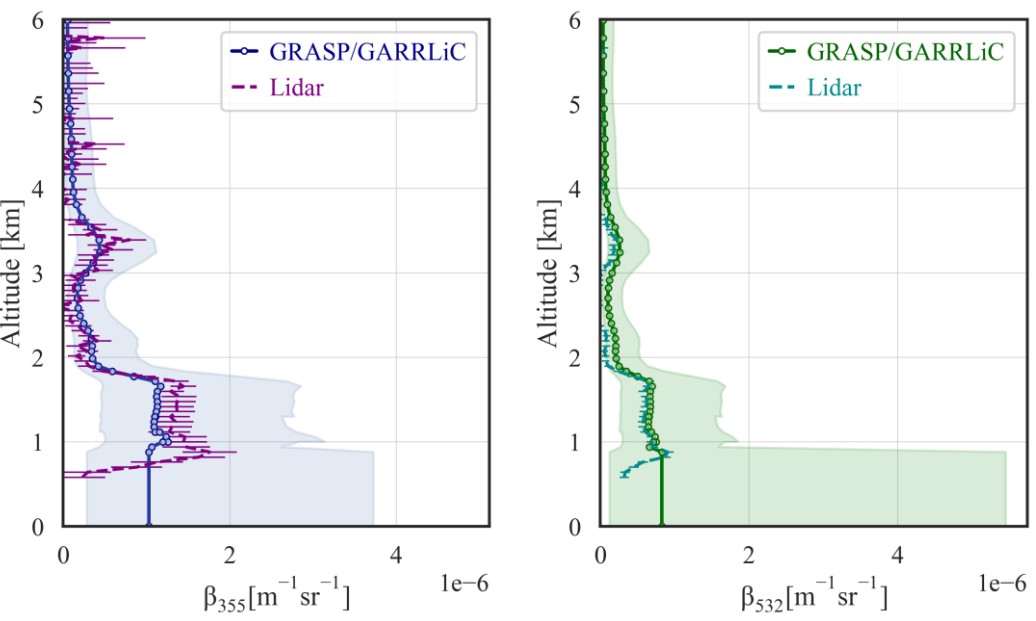





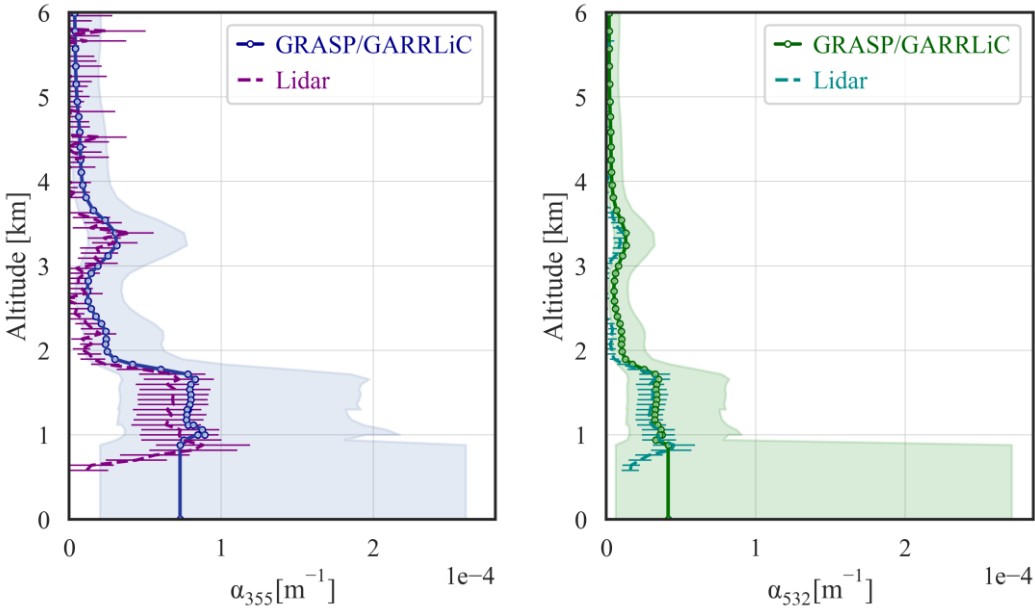

**Figure 10: Evaluation of the backscatter (upper row) and extinction coefficient profiles (lower row) at 355 and 532nm, derived from GRASP/GARRLiC (solid lines with circles), against the SCC corresponding products (dashed lines), for the case study from Antikythera EARLINET station on 25 May 2020, 13:15 - 15:00 UTC. Altitude is provided in km a.s.l.**











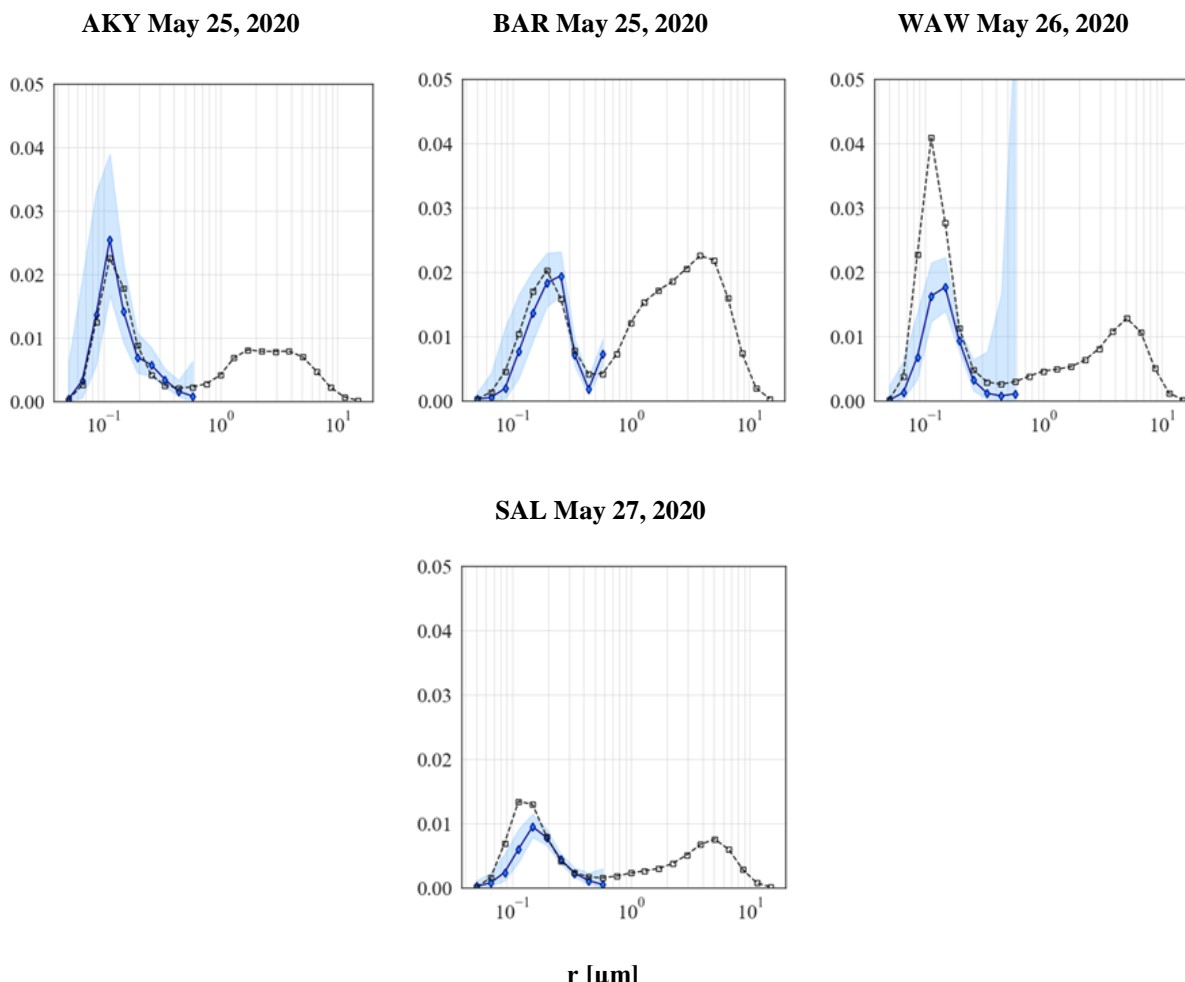

Figure 11: The retrieved volume size distributions of the anthropogenic component over Europe (blue line), during the lockdown and relaxation period of May 2020 due to the COVID-19 pandemic. The shaded blue area represents the GRASP/GARRLiC retrieval uncertainty for the volume size distribution. The AERONET size distribution used for the evaluation of the results is also provided for each case (black line with squares).

**COG May 4, 2020**

**PAY May 6, 2020**

**SAL May 7, 2020**

**BAR May 18, 2020**

**PAY May 20, 2020**

**WAW May 22, 2020**

**AKY May 23, 2020**

**KUO May 23, 2020**

**KUO May 24, 2020**

Altitude [km]



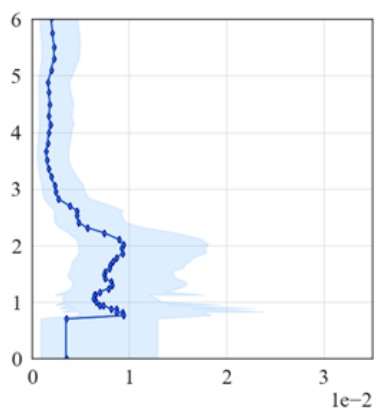

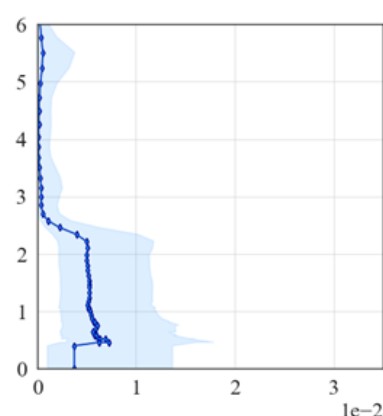

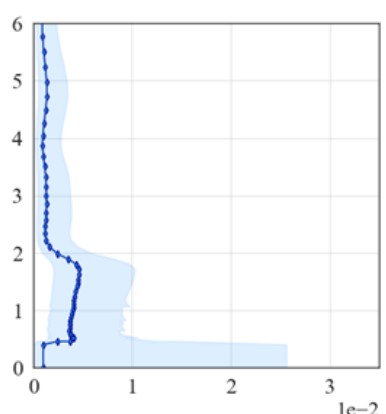

**AKY May 25, 2020**         **BAR May 25, 2020**         **WAW May 26, 2020**

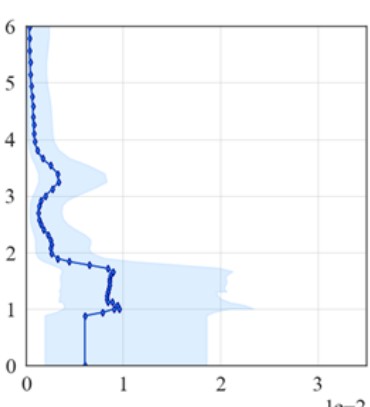

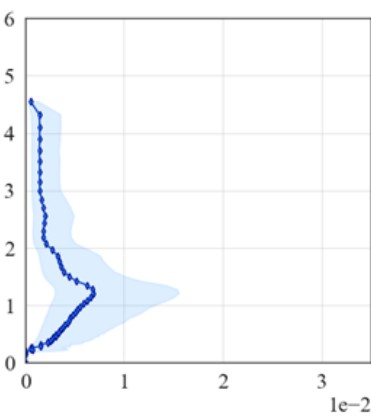

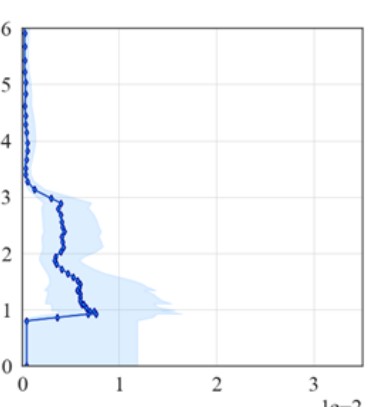

**SAL May 27, 2020**

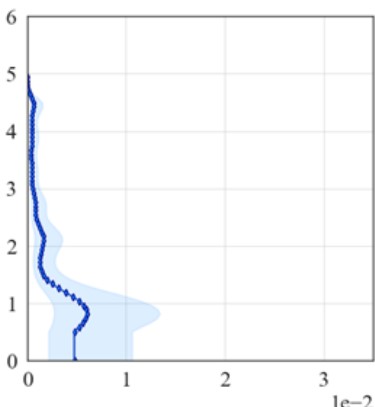



**Concentration [ppb]**

**Figure 12: The retrieved concentration profiles of the anthropogenic component over Europe (blue line), during the lockdown and relaxation period of May 2020 due to the COVID-19 pandemic. Altitude is provided in km a.s.l. The shaded blue area represents the retrieval uncertainty of GRASP/GARRLiC for the concentration profiles.**

925

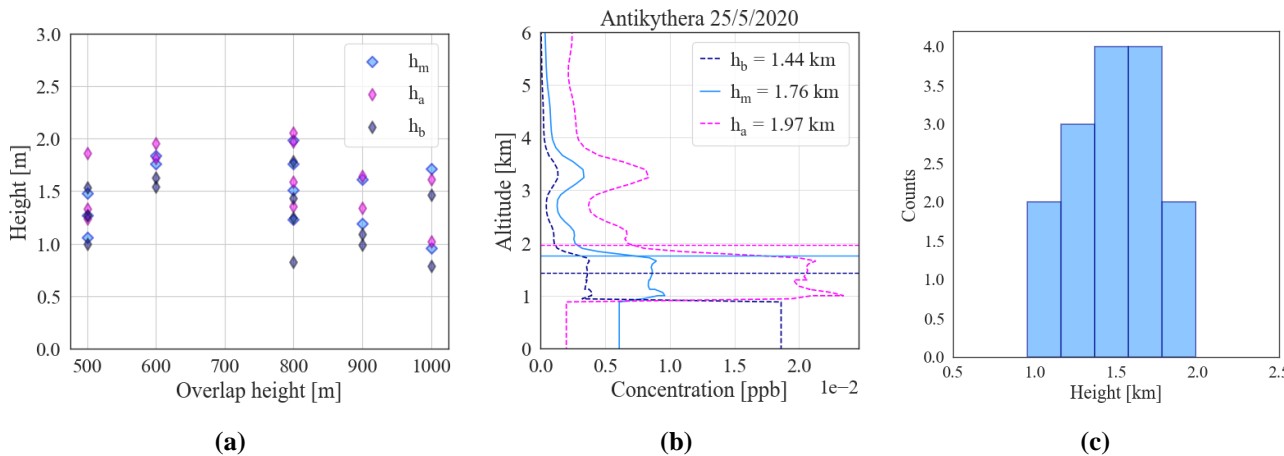

(a)      (b)      (c)

**Figure 13: a) Scatter plot of the center-of-mass (COM) of the anthropogenic component, as derived from GRASP/GARRLiC concentration profile, with respect to the full overlap height of the different lidar systems. "$h_m$" (blue) represents the COM as derived from the concentration profile without accounting of the retrieval uncertainty, "$h_a$" (pink) represents the COM as derived from the concentration profile taking into account the maximum value above the full overlap height and the minimum value below the full overlap height, and "$h_b$" (dark blue) represents the COM as derived from the concentration profile taking into account the minimum value above the full overlap height and the maximum value below the full overlap height. Detailed calculations of "$h_m$", "$h_a$" and "$h_b$" for each case study are included in the Supplement. b) Example of "$h_m$" (blue), "$h_a$" (pink) and "$h_b$" (dark blue) profiles, for the case study from Antikythera EARLINET station on 25 May 2020. Altitude is provided in km a.s.l. c) Histogram of $h_m$-COM values of the anthropogenic component concentration profile for all cases studied herein.**

930

935



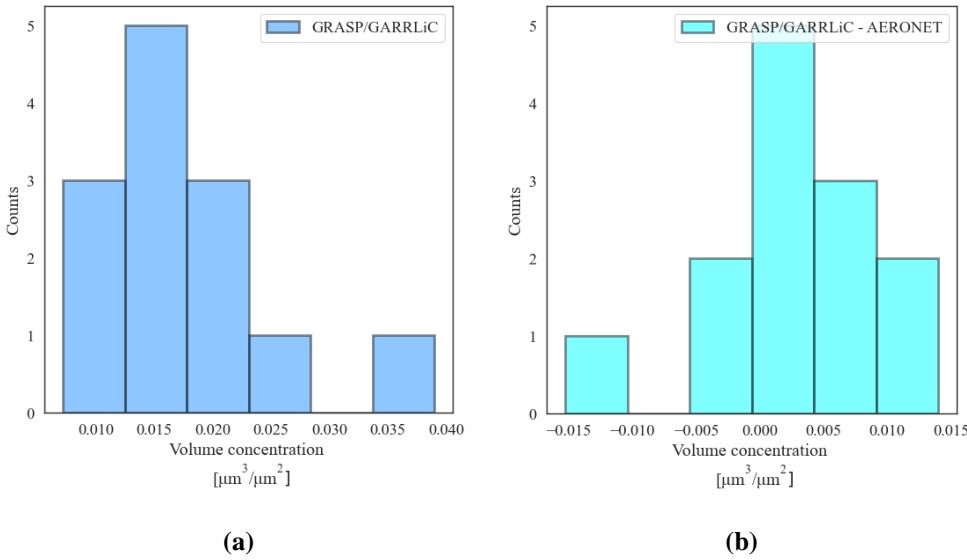

(a)        (b)

**Figure 14: a) Histogram of the volume concentration of the anthropogenic aerosol component, as retrieved from GRASP/GARRLiC.**
940 **b) Histogram of the absolute difference of (a) with the volume concentration of the fine mode from AERONET.**

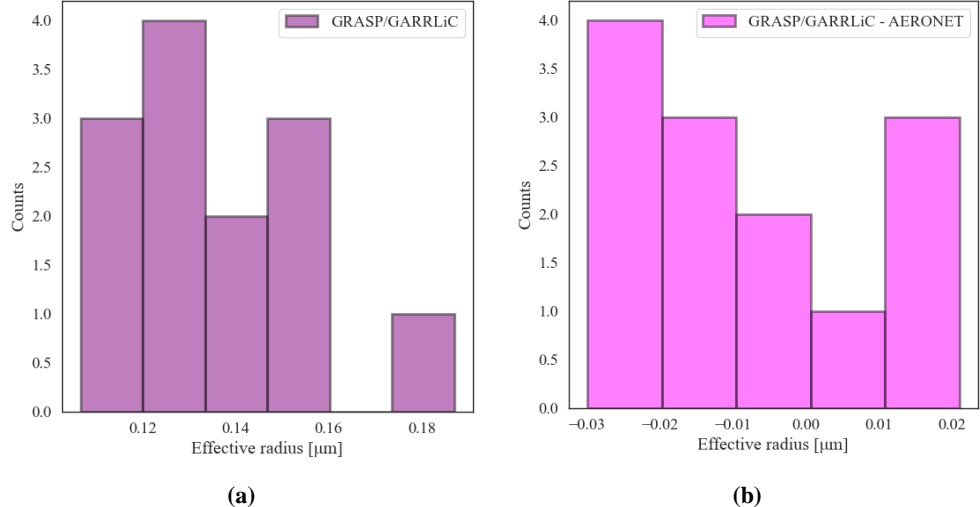

(a)        (b)

**Figure 15: a) Histogram of the mean effective radius of the anthropogenic aerosol component, as retrieved from GRASP/GARRLiC.**
945 **b) Histogram of the absolute difference of (a) with the effective radius of the fine mode from AERONET.**