# Peer review of "Combined sun-photometer/lidar inversion: lessons learned during the EARLINET/ACTRIS COVID-19 Campaign"

_Atmospheric Measurement Techniques, 2023_

## Referee Comment (RC1)

Review of "Combined sun-photometer/lidar inversion: lessons learned during the EARLINET/ACTRIS COVID-19 Campaign" by Tsekeri et al.

This article discusses the use of the GRASP/GARRLIC algorithm applied to lidar and AERONET Sun Photometer measurements acquired during May 2020 over Europe. These measurements, acquired by sensors in the ACTRIS network, were used to derive profiles of fine mode aerosol size distribution and concentration for selected cases that considered to be dominated by anthropogenic aerosols. The original intent of the paper was to monitor the spatial and temporal characteristics of anthropogenic aerosols during the COVID lockdown to study the impacts of this lockdown on these aerosol properties. However, the lack of suitable conditions and other constraints on the lidar measurements prevented the acquisition of sufficient data to address these objectives. Instead, the article focuses on the methodology used to implement the algorithm and the results of the retrievals for a few cases. The paper describes the selection of cases that to be dominated by anthropogenic aerosols, describes the lidar and Sun photometer that were selected and used in the retrieval algorithm, the retrieval algorithm methodology, comparison of retrieval results with other lidar and Sun photometer measurements/retrievals, and issues to be addressed in the use of the algorithm and the interpretation of the results. The paper is generally well written.

As stated above the paper focuses on the methodology of the GRASP/GARRLIC algorithm and the application of this algorithm to retrieving fine mode aerosol size distribution and profiles of fine mode aerosol concentration and the assumed to be anthropogenic. This intent was for this methodology to be application to data acquired by similar lidars and Sun photometers in this network. However, in practice, this methodology appears to have limitations that appear to restrict its use. More importantly, the actual implementation of the algorithm is not clearly presented as described in the comments below. I would not recommend publication until and unless the authors adequately address the comments below.

1. The paper indicates the purpose of the algorithm is to focus on the retrieval of fine mode anthropogenic aerosols. Line 159 indicates that cases with fine mode natural aerosols were filtered out and line 109 indicates that back trajectory and emission sensitivity analyses were used to *verify* the absence of fine mode dust or smoke particles. This word *verify* implies that other means, such as the lidar data, were the primary method used to identify and screen out natural aerosols. However, line 220 seems to indicate that these model simulations were the primary means to exclude natural aerosols. Therefore, it would be helpful to clarify exactly how the cases of anthropogenic aerosols were identified. Furthermore, section 3.1 indicates that the lidar depolarization measurements were used to screen out cases of dust aerosols. However, line 91 indicates that fine mode dust has low depolarization; if that is the case, how were cases with fine mode dust identified and removed? How were cases with fine mode aerosols produced by biomass burning (i.e., smoke) identified and

removed? Was it assumed that if the backtrajectories did not appear to come from regions of fires, that the aerosols were not comprised of smoke?

2. Section 3.4 indicates that a-priori knowledge in the form of smoothing constraints is required for these aerosol retrievals. Section 4.1 describes the optimization of the "setting" or smoothing parameters used by the GRASP/GARRLIC algorithm as well as the sensitivity of the algorithm to the initial guess of the input parameters. It is not clear how the initial guesses of the input parameters are determined; likewise, it is not clear how these smoothing constraints are determined and how the retrieval results depend on these smoothing constraints. What are the ranges of the parameters that are input? In addition, it appears that the "optimum" retrieval is selected from several acceptable solutions based on the least amount of smoothing and the absence of unphysical oscillations in the retrieved parameters. Since lines 276 sand 300 indicate that the latter criterion is based on a qualitative and subjective manner, it seem that this would make it difficult to implement over a network of lidars in a consistent manner. Also, if the acceptable solutions depend on the initial guesses, how does one know what range of initial guesses is acceptable, so that the solutions are acceptable? How does one separate the process of determining the appropriate smoothing parameters from the process of determining the appropriate range of the initial guesses of the input parameters?

3. Speaking of smoothing parameters, section 4.1.1 indicates that these smoothing parameters may vary for different atmospheric states. Why? It is also stated that a global optimum combination of setting parameters was not possible, due in part to the low (<0.15 fine mode AOD at 500 nm.) What is meant by different atmospheric states? Does this mean different AOD amounts? Different fine mode size distributions? Different fine mode aerosol compositions? This seems to relate to the initial guesses for the input parameters discussed above. How can a robust retrieval method be implemented if it depends on the atmospheric state?

4. Lines 311-333. Here it is stated that the information content in the lidar measurements is extremely low relative to the information content in the sky radiances from the Sun photometer so that the addition of the lidar data is not expected to have a strong impact on the derived aerosol microphysical properties. However, that is apparently contradicted by the statements in lines 380-384, which suggest that the difference in the retrieved effective radius compared to the standard AERONET retrievals is because of the higher information content in the lidar measurements, particularly at 355 nm. Has there been any study done to examine the extent to which the lidar measurements provide additional aerosol size distribution information above the sky radiance measurements?

5. Section 4.2 and Figure 10 discuss and show the use of the SCC backscatter and extinction profiles to help evaluate the corresponding profiles calculated from the GRASP/GARRLIC retrieved aerosol properties. Line 335 indicates that these SCC profiles were computed using a pre-defined lidar ratio characteristic for each scene. How was

this lidar ratio determined for each scene? What is the uncertainty in this lidar ratio and how would this uncertainty impact the use of these SCC profiles for evaluation of the GRASP/GARRLIC backscatter and extinction profiles?

Other Comments:

1. Line 63. "…comprised of thirty-three permanent…" instead of "…comprised by…"
2. Line 78. This discussion should be modified to indicate that ground-based lidar detection suffers from these overlap constraints. Airborne and satellite lidars do not have these problems when measuring near-surface aerosols.
3. Line 166. A laser repetition rate of 20 Hz is actually a low rep rate; there are lidars that use lasers with kHz rep rates.
4. Section 3.4 indicates that the retrieval output includes the columnar aerosol volume size distribution, and that the algorithm currently assumes a fixed atmospheric profile. It sounds like the fine mode (total?) aerosol size distribution (and aerosol composition?) are assumed to be constant with altitude. Is this correct? If so, this needs to be clearly stated. Were the lidar multiwavelength profile measurements of aerosol backscatter (and aerosol extinction if available) examined and used to identify cases that may satisfy this assumption? What is done to account for cases where the aerosol backscatter profiles increase with increased relative humidity causing a change in aerosol size but not necessarily concentration?
5. Related to the previous question, section 6.2 discusses the assumption of a homogeneous layer within the lidar overlap region. Since a significant fraction of the column aerosol optical thickness and concentration can reside in this region, this assumption should be mentioned earlier in the discussion.
6. Line 290 and Figure 3a. Should be rectangle instead of rectangular.
7. Line 388. It is suggested that the difference in the aerosol size distribution retrieved using the GRASP/GARRLIC method and the standard AERONET retrieval method may be due to an incorrect estimation of the molecular scattering contribution. Were different density models used to determine the sensitivity of the retrieved aerosol size distribution to the model density to test this hypothesis?
8. Line 339 is missing some references.
9. Line 400. The selection of suitable cases to combine the lidar and Sun Photometer measurements used spatial and temporal collocation criteria of maximum of 30 min and 1 km. These criteria appear inconsistent. Even with low (~3 m/s) wind speeds, one would expect aerosols to travel several kilometers during a 30-minute period.
10. Table 4 lists several different smoothing constraints including those for real and imaginary refractive index. Does the retrieval technique rely on a-priori assumptions of refractive indices, and if so, how are these determined?

11. Figure 5.  What are the units for the parameters plotted on the x and y axes in each of these graphs?

---

## Author Comment (AC1)

On behalf of all co-authors of this paper, I would like to thank you a lot for your time to evaluate our manuscript and for your valuable feedback. Below you may find our replies to your comments written with blue colour. New or changed sentences/paragraphs in our manuscript are also provided (in black colour/italics).

**A. Major comments**

1.      The paper indicates the purpose of the algorithm is to focus on the retrieval of fine mode anthropogenic aerosols. Line 159 indicates that cases with fine mode natural aerosols were filtered out and line 109 indicates that back trajectory and emission sensitivity analyses were used to *verify* the absence of fine mode dust or smoke particles. This word *verify* implies that other means, such as the lidar data, were the primary method used to identify and screen out natural aerosols. However, line 220 seems to indicate that these model simulations were the primary means to exclude natural aerosols. Therefore, it would be helpful to clarify exactly how the cases of anthropogenic aerosols were identified. Furthermore, section 3.1 indicates that the lidar depolarization measurements were used to screen out cases of dust aerosols. However, line 91 indicates that fine mode dust has low depolarization; if that is the case, how were cases with fine mode dust identified and removed? How were cases with fine mode aerosols produced by biomass burning (i.e., smoke) identified and removed? Was it assumed that if the back-trajectories did not appear to come from regions of fires, that the aerosols were not comprised of smoke?

**REPLY:** We thank the reviewer for this comment that help us to better explain the methodology applied in order to select atmospheric scenes excluding the presence of natural aerosols. The process is as follows: first, the scenes with particle linear depolarization ratio (PLDR) greater than a few percent were rejected. This selection is based on earlier studies that utilize polarization lidar data and conclude that PLDR for anthropogenic aerosols is not expected to exceed ~5% at both 355 and 532 nm (i.e., Muller et al., 2007; Preibler et al., 2013; Heese et al., 2015; Illingworth et al., 2015; Giannakaki et al., 2016; Kaduk et al., 2017). Values higher than that indicate a possible mixture of the particles with highly depolarizing aerosols such as desert dust (i.e., Preibler et al., 2013; Janicka et al., 2016; Kaduk et al., 2017; Rittmeister et al., 2017; Floutsi et al., 2023). Pure coarse mode dust particles can induce PLDR of the order of 30 – 40% at 532 nm (i.e., Freudenthaler et al., 2009; Sakai et al., 2010; Burton et al., 2015; Veselovskii et al., 2016). For fine mode dust, based on laboratory studies, PLDR values can be lower and of the order of 14 – 17% (Sakai et al., 2010; Jarvinen et al., 2016), which however is still high compared to anthropogenic particles.

Regarding the exclusion of cases with biomass burning aerosols: Lidar-derived optical properties of biomass burning and anthropogenic aerosols can be quite similar, although the former present higher lidar ratio (LR) values, which indicate the highly-absorbing nature of the particles in contrast to anthropogenic particles. For example, in the recent study of Floutsi et al. (2023), the reported mean LR values for biomass burning and anthropogenic pollution particles are $68.2 \pm 7.4$ and $71.8 \pm 11.1$ sr at 355 nm and $51.1 \pm 8.7$ and $47.4 \pm 7.4$ sr at 532 nm, respectively

(see also Fig. 1). These results are derived mainly from European routine multi-wavelength Raman lidar measurements but also worldwide campaigns.

In the context of our study, in order to verify the absence of biomass burning aerosols, a combination of FLEXPART Lagrangian transport model runs and satellite images was utilized: In particular, the origin regions of the air masses derived using FLEXPART back-trajectories were examined together with the 'Fires and Thermal Anomalies' product from VIIRS instrument on board Suomi-NPP. The cases for which the area of origin of the air masses seemed to be affected by the presence of active fires, were rejected.

[Figure]

[Figure]

**Figure 1)** Intensive optical properties (PLDR and LR) of various aerosol types as derived from routine and campaign multi-wavelength, polarization, Raman lidar measurements in worldwide locations. (top) LR versus the PLDR at 355 nm, (bottom) the same at 532 nm. Measurements that refer to biomass burning and anthropogenic pollution aerosols are highlighted in red and green circles, respectively. (Source: Floutsi et al., 2023)

We appreciate that the selection process and filtering of the atmospheric scenes might not be very clear to the reader in the present manuscript version, thus the following manuscript parts have been modified accordingly:

**Lines 91 – 92:** "*Although fine dust presents lower depolarization values (Järvinen et al., 2016; Sakai et al., 2010; Szczepanik et al. 2021), they are still higher than the depolarization values of anthropogenic particles.*"

**Lines 108 – 111:** "*Herein, we use a similar synergistic approach, in order to derive the microphysical properties of the anthropogenic aerosol component in different European areas. First, we ensure the absence of natural aerosols in the atmospheric scenes: For excluding cases with transported (fine) dust particles, we use lidar depolarization measurements along with air-mass back-trajectory and emission sensitivity analysis. For excluding smoke cases, we utilize a combination of model runs and satellite images, in order to identify air masses that were affected by the presence of active fires.*"

**Lines 160-162:** "*For excluding cases with transported (fine) dust particles, we used the modelling tools described in Sec. 3.3, along with lidar measurements of the volume and particle linear depolarization ratio (VLDR and PLDR, respectively), indicating non-spherical particles (i.e., dust particles), as discussed in Sect. 3.1. For excluding smoke cases, we identified air masses that were affected by the presence of active fires, by examining the back-trajectories of the air-masses (Sec. 3.3) in combination with the 'Fires and Thermal Anomalies' product from VIIRS instrument on board Suomi-NPP satellite.*"

**Lines 219 – 221:** "*In order to identify the aerosol sources during the EARLINET/ACTRIS COVID-19 campaign, we used atmospheric model simulations that helped us, along with lidar measurements, to exclude fine natural aerosol particles (dust and smoke), and focus only on the anthropogenic aerosol component. This was done through the identification of the origin of the aerosol particles.*"

**References**

- Burton, S. P., Hair, J. W., Kahnert, M., Ferrare, R. A., Hostetler, C. A., Cook, A. L., Harper, D. B., Berkoff, T. A., Seaman, S. T., Collins, J. E., Fenn, M. A., and Rogers, R. R.: Observations of the spectral dependence of linear particle depolarization ratio of aerosols using NASA Langley airborne High Spectral Resolution Lidar, Atmos. Chem. Phys., 15, 13453–13473, https://doi.org/10.5194/acp-15-13453-2015, 2015.

- Floutsi, A. A., Baars, H., Engelmann, R., Althausen, D., Ansmann, A., Bohlmann, S., Heese, B., Hofer, J., Kanitz, T., Haarig, M., Ohneiser, K., Radenz, M., Seifert, P., Skupin, A., Yin, Z., Abdullaev, S. F., Komppula, M., Filioglou, M., Giannakaki, E., Stachlewska, I. S., Janicka, L., Bortoli, D., Marinou, E., Amiridis, V., Gialitaki, A., Mamouri, R.-E., Barja, B., and Wandinger, U.: DeLiAn – a growing collection of depolarization ratio, lidar ratio and Ångström exponent for different aerosol types and mixtures from ground-based lidar observations, Atmos. Meas. Tech., 16, 2353–2379, https://doi.org/10.5194/amt-16-2353-2023, 2023.

- Freudenthaler, V., Esselborn, M., Wiegner, M., Heese, B., Tesche, M., Ansmann, A., Müller, D., Althausen, D., Wirth, M., Fix, A., Ehret, G., Knippertz, P., Toledano, C., Gasteiger, J., Garhammer, M., and Seefeldner, M.: Depolarization ratio profiling at several wavelengths in pure Saharan dust during SAMUM 2006, Tellus B, 61, 165–179, https://doi.org/10.1111/j.1600-0889.2008.00396.x, 2009.

- Giannakaki, E., van Zyl, P. G., Müller, D., Balis, D., and Komppula, M.: Optical and microphysical characterization of aerosol layers over South Africa by means of multi-wavelength depolarization and Raman lidar measurements, Atmos. Chem. Phys., 16, 8109–8123, https://doi.org/10.5194/acp-16-8109-2016, 2016.

- Illingworth, A. J., Barker, H. W., Beljaars, A., Ceccaldi, M., Chepfer, H., Clerbaux, N., Cole, J., Delanoe, J., Domenech, C., Donovan, D. P., Fukuda, S., Hirakata, M., Hogan, R. J., Huenerbein, A., Kollias, P., Kubota, T., Nakajima, T., Nakajima, T. Y., Nishizawa, T., Ohno, Y., Okamoto, H., Oki, R., Sato, K., Satoh, M., Shephard, M. W., Velazquez-Blazquez, A., Wandinger, U., Wehr, T., and van Zadelhoff, G. J.: THE

EARTHCARE SATELLITE The Next Step Forward in Global Measurements of Clouds, Aerosols, Precipitation, and Radiation, B. Am. Meteorol. Soc., 96, 1311–1332, https://doi.org/10.1175/bams-d-12-00227.1, 2015.

- Janicka, L., Stachlewska, I. S., Markowicz, K. M., Baars, H., Engelmann, R., and Heese, B.: Lidar Measurements of Canadian Forest Fire Smoke Episode Observed in July 2013 over Warsaw, Poland, Proceedings of the 27th International Laser Radar Conference (ILRC 27), 5–10 July 2015, New York City, USA, EPJ Web Conf., 119, 18005, https://doi.org/10.1051/epjconf/201611918005, 2016.

- Järvinen, E., Kemppinen, O., Nousiainen, Kociok, T., Möhler, O., Leisner, T., and Schnaiter, M.: Laboratory investigations of mineral dust near-backscattering depolarization ratios, J. Quant. Spectrosc. Ra., 178, 192–208, https://doi.org/10.1016/j.jqsrt.2016.02.003.

- Kaduk, C.: Characterization of the optical properties of complex aerosol mixtures observed with a multiwavelength–Raman–polarization lidar during the 6-weeks BACCHUS campaign in Cyprus in spring 2015, MSc thesis, Leipzig University, 2017.

- Müller, D., Ansmann, A., Mattis, I., Tesche, M., Wandinger, U., Althausen, D., and Pisani, G.: Aerosol-type-dependent lidar ratios observed with Raman lidar, J. Geophys. Res.-Atmos., 112, D16202, https://doi.org/10.1029/2006jd008292, 2007.

- Preißler, J., Wagner, F., Guerrero-Rascado, J. L., and Silva, A. M.: Two years of free-tropospheric aerosol layers observed over Portugal by lidar, J. Geophys. Res.-Atmos., 118, 3676–3686, https://doi.org/10.1002/jgrd.50350, 2013.

- Rittmeister, F., Ansmann, A., Engelmann, R., Skupin, A., Baars, H., Kanitz, T., and Kinne, S.: Profiling of Saharan dust from the Caribbean to western Africa – Part 1: Layering structures and optical properties from shipborne polarization/Raman lidar observations, Atmos. Chem. Phys., 17, 12963–12983, https://doi.org/10.5194/acp-17-12963-2017, 2017.

- Sakai, T., Nagai, T., Zaizen, Y, and Mano, Y.: Backscattering linear depolarization ratio measurements of mineral, sea-salt, and ammonium sulfate particles simulated in a laboratory chamber, Appl. Opt., 49, 4441–4449, 2010.

- Veselovskii, I., Goloub, P., Podvin, T., Bovchaliuk, V., Derimian, Y., Augustin, P., Fourmentin, M., Tanré, D., Korenskiy, M., Whiteman, D. N., Diallo, A., Ndiaye, T., Kolgotin, A., and Dubovik, O.: Retrieval of optical and physical properties of African dust from multiwavelength Raman lidar measurements during the SHADOW campaign in Senegal, Atmos. Chem. Phys., 16, 7013– 7028, https://doi.org/10.5194/acp-16-7013-2016, 2016.

2. Section 3.4 indicates that a-priori knowledge in the form of smoothing constraints is required for these aerosol retrievals. Section 4.1 describes the optimization of the "setting" or smoothing parameters used by the GRASP/GARRLIC algorithm as well as the sensitivity of the algorithm to the initial guess of the input

parameters. It is not clear how the initial guesses of the input parameters are determined; likewise, it is not clear how these smoothing constraints are determined and how the retrieval results depend on these smoothing constraints. What are the ranges of the parameters that are input? In addition, it appears that the "optimum" retrieval is selected from several acceptable solutions based on the least amount of smoothing and the absence of unphysical oscillations in the retrieved parameters. Since lines 276 and 300 indicate that the latter criterion is based on a qualitative and subjective manner, it seems that this would make it difficult to implement over a network of lidars in a consistent manner. Also, if the acceptable solutions depend on the initial guesses, how does one know what range of initial guesses is acceptable, so that the solutions are acceptable? How does one separate the process of determining the appropriate smoothing parameters from the process of determining the appropriate range of the initial guesses of the input parameters?

**REPLY:** We thank the reviewer for this comment. Let us clarify first that we do not "determine the appropriate range of the initial guesses", we rather vary the initial guesses so as to investigate the sensitivity of the optimum solution on this variation. The process of determining the appropriate smoothing parameters is separated from the process of varying the initial guesses. More specifically, the scope for selecting the appropriate smoothing constraints is to derive the optimum retrieval, whereas the scope for varying the initial guesses is to investigate the sensitivity of this optimum retrieval to the initial guess (providing a measure of the robustness of the optimum retrieval).

That being said, it was found that the optimum retrieval did not present a strong dependence on the initial guess since most of the solutions derived with variable initial guess were confined within the retrieval uncertainty.

Regarding the setting of the smoothing constraints: The application of smoothing constraints is an established technique that has been demonstrated in various studies to satisfactory eliminate unrealistic oscillations in the retrievals (Tiknonov and Arsenin 1977; Dubovik and King 2000). In order to determine the optimum smoothing constrains for each case study, we used the range of values shown in Table 4 in the manuscript. This range is determined empirically for the specific cases shown in our study, so as the retrieved properties present various degrees of smoothing. The dependence of the retrieved results on the smoothing parameters was not investigated further than selecting the least smoothing possible, which in the same time provides enough smoothing of unrealistic oscillations for the retrieval.

Regarding the application of this methodology at a network of lidars in a consistent manner, our results are only indicative, due to the very low aerosol optical depth (<0.2 in most cases) that humpers the retrieval of some parameters such as the complex refractive index, and the small number of available case studies. Current efforts from AERIS/ICARE have not concluded as of yet either, as stated in the manuscript. However, the implementation of GRASP on a global scale (i.e., in multi-year record of POLDER measurements) has shown that the application

of common, not-location-related smoothing constrains is possible particularly when accounting for multiple pixels to be inverted simultaneously (i.e., see Dubovik et al., 2011).

Regarding the varying of the initial guesses: As stated above, the scope for varying the initial guesses is to investigate the sensitivity of the optimum retrieval to the initial guess, providing a measure of the robustness of the optimum retrieval. The range of values used for this variation is the same for all cases investigated in our work, it is defined by the minimum/maximum values allowed for the retrieved parameters in GRASP/GARRLiC, and is provided in Table 1 below (also included in the revised version of the Supplement as Table S2). The range of values used cover the physically-expected range of values for the parameters of atmospheric aerosols (e.g., Dubovik et al., 2002).

**Table 1)** Minimum and maximum values of the retrieved parameters in GRASP/GARRLiC. These values provide also the range within which the initial guess for each retrieved parameter is allowed to randomly vary.

| Retrieved parameter | min | max |
|---|---|---|
| Size distribution bins (fine mode) | 0.000005 | 0.5 |
| Size distribution bins (coarse mode) | 0.000005 | 1.0 |
| Real part of the refractive index (*) | 1.33 | 1.6 |
| Imaginary part of the refractive index (*) | 0.0005 | 0.01 |
| Sphericity fraction (coarse mode) | 0.00001 | 0.9999 |
| Vertical profile | 0.000001 | 0.02 |

(*) same limits apply to both fine and coarse mode

The following manuscript parts have been modified accordingly:

**Lines 247 – 249:** "*The methodology for applying the GRASP/GARRLiC algorithm on a network level is based on a two-step approach, first optimizing the parameters used to run the GRASP/GARRLiC retrieval and derive the "optimum retrieval" (Section 4.1.1), and then evaluating the robustness of the optimum retrieval through evaluating its sensitivity to the initial guess (Section 4.1.2).*"

**Line 250:** "*4.1.1 First step: optimizing the setting parameters of GRASP/GARRLiC retrieval and deriving the "optimum retrieval"*"

**Line 264:** "*The range of values used is determined empirically for the specific cases, so as the retrieved properties present various degrees of smoothing.*"

**Line 308:** "*4.1.2 Second step: sensitivity of the GRASP/GARRLiC optimum retrieval to initial guess*"

**Line 313:** "*The range of values used for the initial guess of the different aerosol parameters (Table S2 in supplement) cover the physically-expected range of values for atmospheric aerosols (e.g. Dubovik et al., 2002).*"

**References:**

- Dubovik, O. and King, M. D.: A flexible inversion algorithm for retrieval of aerosol optical properties from Sun and sky radiance measurements, J. Geophys. Res., 105, 20673–20696, 2000.

- Dubovik, O., Holben, B., Eck, T., Smirnov, A., Kaufman, Y., King, M., Tanré, D., and Slutsker, I.: Variability of absorption and optical properties of key aerosol types observed in worldwide locations, J. Atmos. Sci., 59, 590–608, 2002.

- Dubovik, O., Herman, M., Holdak, A., Lapyonok, T., Tanré, D., Deuzé, J. L., Ducos, F., Sinyuk, A., and Lopatin, A.: Statistically optimized inversion algorithm for enhanced retrieval of aerosol properties from spectral multi-angle polarimetric satellite observations, Atmos. Meas. Tech., 4, 975–1018, https://doi.org/10.5194/amt-4-975-2011, 2011.

- King, M. D.: Sensitivity of constrained linear inversions to the selection of the Lagrange multiplier, J. Atmos. Sci., 39, 1356– 1369, 1982.

- Tikhonov, A. N. and Arsenin, V. Y.: Solution of Ill-Posed Problems, Wiley, New York, 300 pp., 1977

- Twomey, S.: Introduction to the Mathematics of Inversion in Remote Sensing and Indirect Measurements, Elsevier, New York, 1977

3. Speaking of smoothing parameters, section 4.1.1 indicates that these smoothing parameters may vary for different atmospheric states. Why? It is also stated that a global optimum combination of setting parameters was not possible, due in part to the low (<0.15 fine mode AOD at 500 nm.) What is meant by different atmospheric states? Does this mean different AOD amounts? Different fine mode size distributions? Different fine mode aerosol compositions? This seems to relate to the initial guesses for the input parameters discussed above. How can a robust retrieval method be implemented if it depends on the atmospheric state?

**REPLY:** We thank the reviewer for this comment.

By different atmospheric states we mean different AOD and different aerosol types (with different size distributions, refractive indices etc). As we state in the manuscript (lines 259-261), the investigation of a "global" optimum combination of the setting parameters that could be used for all cases, was not conclusive (in part due to the low AOD of the case studies), thus we cannot answer to the question why we see variation of the optimum smoothing parameters for different atmospheric states. The only hind we can provide is that the low AOD has as a result a less robust retrieval, which is not optimum to be used to derive "global" estimations for the setting parameters.

As we clarify in the previous comment, the variation of the initial guesses is done after the setting of the smoothing parameters, (in order to provide a measure of the robustness of the optimum retrieval) and it does not affect the setting of smoothing parameters.

The following manuscript part has been modified accordingly:

**Lines 260-261:** "*This analysis was not conclusive, in part due to the low AOD of the case studies available in the current analysis (with AOD values lower than 0.15 at 500 nm for fine particles), resulting in less robust retrievals.*"

4.      Lines 311-333. Here it is stated that the information content in the lidar measurements is extremely low relative to the information content in the sky radiances from the Sun photometer so that the addition of the lidar data is not expected to have a strong impact on the derived aerosol microphysical properties. However, that is apparently contradicted by the statements in lines 380-384, which suggest that the difference in the retrieved effective radius compared to the standard AERONET retrievals is because of the higher information content in the lidar measurements, particularly at 355 nm. Has there been any study done to examine the extent to which the lidar measurements provide additional aerosol size distribution information above the sky radiance measurements?

**REPLY:** We thank the reviewer for this comment.

As stated in the manuscript, the addition of the lidar data is not expected to have a strong impact on the derived aerosol size distribution, but this does not mean that it is not expected to have any impact. Thus, the difference in the retrieved effective radius compared to the standard AERONET retrieval found for some of the case studies may be attributed to the additional information content in the lidar measurements at 355 nm. In the manuscript (lines 384-388) we provide another possible reason for this difference: it may be also attributed in the way the molecular contribution to the signal is represented in the forward model.

A study on the benefits of the joint inversion of lidar and sun-photometer data with GRASP/GARRLiC is provided by Lopatin et al. (2013).

The following manuscript part has been modified to include the above:

**Lines 330-333:** "*In principle, the information content of lidar measurements on the aerosol size distribution is low compared to the one contained in the sky radiances measured from the sky/sun-photometer. Therefore, we do not expect the addition of the lidar measurements in the retrieval to have a strong impact on the derived size distribution (this does not mean though that we expect no impact, as discussed in Sec. 5).*"

**References:**

- Lopatin, A., Dubovik, O., Chaikovsky, A., Goloub, P., Lapyonok, T., Tanré, D., and Litvinov, P.: Enhancement of aerosol characterization using synergy of lidar and sun-photometer coincident observations: the GARRLiC algorithm, Atmos. Meas. Tech., 6, 2065–2088, https://doi.org/10.5194/amt-6-2065-2013, 2013.

5.      Section 4.2 and Figure 10 discuss and show the use of the SCC backscatter and extinction profiles to help evaluate the corresponding profiles calculated from the GRASP/GARRLIC retrieved aerosol properties. Line 335 indicates that these SCC profiles were computed using a pre-defined lidar ratio characteristic for each scene. How was this lidar ratio determined for each scene? What is the uncertainty in this lidar ratio and how would this uncertainty impact the use of these SCC profiles for evaluation of the GRASP/GARRLIC backscatter and extinction profiles?

**REPLY:** We thank the reviewer for this question.

In the context of our study, the requirement for simultaneous measurements (less than +/-30 min difference) between the lidar and the sun-photometer, imposes the use of elastic lidar signals acquired under strong daylight conditions. Thus, appropriate LR values needed to be selected for each case study, in order to derive the backscatter and extinction coefficient profiles from the lidar measurements alone and use those to evaluate the GRASP/GARRLiC synergistic retrievals.

During the month of the intensive campaign, the EARLINET SCC daytime retrievals were performed assuming fixed LR profiles with values corresponding to each station climatology. For the stations participating in our study and the case studies analysed, these values ranged between 45 and 55 sr with an uncertainty of 10 sr, that covers most of the possible scenarios for different aerosol types and their mixtures in the atmospheric column. Additionally, the range of 35-65 sr covers most of the LR values reported for "pollution" aerosols at 355 and 532nm, as shown in Floutsi et al. (2023) (see Fig. 1 above). The assumed uncertainty of 10 sr is included in the calculations of the backscatter and extinction coefficient uncertainties provided by the SCC retrieval, using error propagation.

We changed lines 334-336 accordingly: "*Since we consider only daytime measurements, the SCC particle extinction coefficient profiles are calculated using the particle backscatter coefficient profiles derived from ELDA and a constant pre-defined lidar ratio (S) value, which is characteristic for each specific scene, ranging between 45 and 55 sr, with an uncertainty of 10 sr. The range of 35-65 sr covers most of the LR values reported for "pollution" aerosols at 355 and 532nm, as shown in Floutsi et al. (2023).*"

**References**

- Floutsi, A. A., Baars, H., Engelmann, R., Althausen, D., Ansmann, A., Bohlmann, S., Heese, B., Hofer, J., Kanitz, T., Haarig, M., Ohneiser, K., Radenz, M., Seifert, P., Skupin, A., Yin, Z., Abdullaev, S. F., Komppula, M., Filioglou, M., Giannakaki, E., Stachlewska, I. S., Janicka, L., Bortoli, D., Marinou, E., Amiridis, V., Gialitaki, A., Mamouri, R.-E., Barja, B., and Wandinger, U.: DeLiAn – a growing collection of depolarization ratio, lidar ratio and Ångström exponent for different aerosol types and mixtures from ground-based lidar observations, Atmos. Meas. Tech., 16, 2353–2379, https://doi.org/10.5194/amt-16-2353-2023, 2023.

**B. Other comments**

1. Line 63. "…comprised of thirty-three permanent…" instead of "…comprised by…"
   **Response:** Thank you, it has been corrected.

2. Line 78. This discussion should be modified to indicate that ground-based lidar detection suffers from these overlap constraints. Airborne and satellite lidars do not have these problems when measuring near-surface aerosols.
   **Response:** Thank you, lines 77-81 have been changed to: "*Lidars are the only instruments that can provide detailed vertically-resolved profiles of aerosol properties (e.g., Ansmann and Müller, 2005). However, in the lowest part of the planetary boundary layer (PBL) (Kotthaus et al. 2023) where most of the anthropogenic aerosols reside, the detection capabilities of ground-based lidars suffer due to the instrument geometry (Chen et al., 2014; Navas-Guzmán et al., 2011; Wandinger and Ansmann, 2002).*"

3. Line 166. A laser repetition rate of 20 Hz is actually a low rep rate; there are lidars that use lasers with kHz rep rates.
   **Response:** Thank you, it has been corrected.

4. Section 3.4 indicates that the retrieval output includes the columnar aerosol volume size distribution, and that the algorithm currently assumes a fixed atmospheric profile. It sounds like the fine mode (total?) aerosol size distribution (and aerosol composition?) are assumed to be constant with altitude. Is this correct? If so, this needs to be clearly stated. Were the lidar multiwavelength profile measurements of aerosol backscatter (and aerosol extinction if available) examined and used to identify cases that may satisfy this assumption? What is done to account for cases where the aerosol backscatter profiles increase with increased relative humidity causing a change in aerosol size but not necessarily concentration?

5. Related to the previous question, section 6.2 discusses the assumption of a homogeneous layer within the lidar overlap region. Since a significant fraction of the column aerosol optical thickness and concentration can reside in this region, this assumption should be mentioned earlier in the discussion.

**Response (provided for both points 4 and 5):** The fine mode size distribution changes with height, since the fine mode concentration changes with height, although the shape of the fine mode size distribution does not change. The fine mode refractive index (i.e. composition) does not change with height. We clarify these points, along with the assumption of a homogeneous layer within the lidar overlap region, in the manuscript (lines 236-238): "*Amongst others, the retrieved parameters include the columnar aerosol total volume size distribution, the columnar spectral complex refractive index at 440, 670, 870 and 1020 nm, and the profiles of aerosol concentration at 60 altitude levels (considering a homogeneous layer of constant concentration from the surface to the full-overlap lidar height), for both fine and coarse aerosol modes (see more details in Lopatin et al. (2013; 2021)). Thus, for the fine mode aerosols investigated here, the size distribution changes with height in terms of their concentration, but its shape is constant with height. Moreover, their spectral refractive index (i.e., composition) does not change with height.*"

The lidar multiwavelength profile measurements of aerosol backscatter and extinction coefficients are used to validate the retrieved aerosol properties (i.e., volume concentration profiles, columnar total size distribution, columnar refractive index), by comparing the GRASP/GARRLiC-derived aerosol backscatter and extinction coefficient profiles with the corresponding products from SCC, as discussed in Sec. 4.2.

The effect of the change of RH humidity with height is not resolved by the GRASP/GARRLiC algorithm for the fine mode aerosols, since the algorithm does not provide vertically-resolved composition of the fine particles, but only their effective-column composition.

6. Line 290 and Figure 3a. Should be rectangle instead of rectangular.

   **Response:** Thank you, it has been corrected.

7. Line 388. It is suggested that the difference in the aerosol size distribution retrieved using the GRASP/GARRLIC method and the standard AERONET retrieval method may be due to an incorrect estimation of the molecular scattering contribution. Were different density models used to determine the sensitivity of the retrieved aerosol size distribution to the model density to test this hypothesis?

   **Response:** Thank you. No such study has been performed. As referred in the manuscript, using a user-provided atmospheric profile is planned to be included in the future in GRASP/GARRLiC.

8. Line 339 is missing some references.

   **Response:** Thank you, it has been corrected.

9. Line 400. The selection of suitable cases to combine the lidar and Sun Photometer measurements used spatial and temporal collocation criteria of maximum of 30 min and 1 km. These criteria appear inconsistent. Even with low (~3 m/s) wind speeds, one would expect aerosols to travel several kilometres during a 30-minute period.

   **Response:** Thank you. The collocation criteria are based on the criteria set from EARLINET for satellite measurement validation (Papagiannopoulos et al., 2016), and as described in lines 400-404 in the manuscript, need to be re-evaluated, since they are expected to depend on the local geographical

characteristics of each site, as well as on the atmospheric conditions: "These fixed threshold selection criteria are based on empirical knowledge of the optimum time and spatial difference between the different measurement datasets (i.e., Papagiannopoulos et al., 2016), and need to be re-evaluated for each station in the network, in order to take into account the effective spatio-temporal variability of the aerosols properties. In principle, the spatio-temporal variability is expected to depend on the local geographical characteristics of the site, as well as on the atmospheric conditions."

10. Table 4 lists several different smoothing constraints including those for real and imaginary refractive index. Does the retrieval technique rely on a-priori assumptions of refractive indices, and if so, how are these determined?

**Response:** Thank you for this comment.

As described in the response for the 2nd major comment above, the range used to determine the optimum smoothing constrains for each case study (shown in Table 4 in the manuscript) is determined empirically for the specific cases shown in our study, so as the retrieved properties present various degrees of smoothing.

The following manuscript part has been modified accordingly:

**Line 264:** "*The range of values used is determined empirically for the specific cases, so as the retrieved properties present various degrees of smoothing.*"

A-priori assumptions for the refractive indices (and also for the rest of the retrieved aerosol parameters) are also considered in terms of minimum and maximum values for the retrieved parameters. These are the same with the minimum and maximum values shown for all parameters in Table 1 above (also included in the revised version of the Supplement as Table S2). These values are derived from the physically-expected range of values for the parameters of atmospheric aerosols (e.g. Dubovik et al., 2002).

The following manuscript part has been modified accordingly:

**Lines 230-231:** "*An example of a-priori knowledge is the minimum and maximum values of the retrieved parameters (derived from the physically-expected range of values for the parameters of atmospheric aerosols (e.g. Dubovik et al., 2002); Table S2 in the Supplement). Another example is the smoothing constraints imposed on the retrieved volume size distributions, or on the spectral variability of the retrieved refractive index.*"

**Line 313:** "*(Note that the minimum and maximum values shown in Table S2 are also used in the retrieval as a-priori assumptions of the minimum and maximum values of the derived parameters.)*"

11. Figure 5. What are the units for the parameters plotted on the x and y axes in each of these graphs?

**Response:** Thank you for this comment.

The parameters plotted in Fig. 5 are a) the total optical depth (unitless), b) the total scattered radiances measured from the sun-photometer and c) the altitude resolved back-scattered radiances measured from the lidar. For GRASP/GARRLiC the radiances used in the inversion are required to be normalized as specified

in https://www.grasp-open.com/doc/ch04.php#sdata-format. Thus, both b and c correspond to normalized values and are hence unitless. We included this information in the plots and caption of Fig. 5.

---

## Author Comment (AC2)

On behalf of all co-authors of this paper, I would like to thank you a lot for your time to evaluate our manuscript and for your valuable feedback. Below you may find our replies to your comments written with blue colour. New or changed sentences/paragraphs in our manuscript are also provided (in black colour/italics).

1. L 56. In this sentence, I suggest replacing "anthropogenic" with "anthropic"

   **Response:** Thank you, it has been corrected.

2. L62-63, "is comprised by" -> consists of

   **Response:** Thank you, it has been corrected.

3. L83, remove "the" before "lidars"

   **Response:** Thank you, it has been corrected.

4. L91. "This is though" -> "However, this is"

   **Response:** Based on Reviewer's #1 comment lines 91-92 have been changed as follows: *"Although fine dust presents lower depolarization values (Järvinen et al., 2016; Sakai et al., 2010; Szczepanik et al. 2021), they are still higher than the depolarization values of anthropogenic particles."*

5. L161. What is the difference between "volume" and "particle" linear depolarization ratio? Can you define these quantities somewhere in the text?

   **Response:** The volume linear depolarization ratio is the linear depolarization ratio taking into account both molecules and aerosol particles in the atmosphere, whereas the particle linear depolarization ratio takes into account only the aerosol particles in the atmosphere. We have included this clarification in line 162: *"(VLDR takes into account both molecules and aerosol particles in the atmosphere, whereas PLDR takes into account only the aerosol particles.)"*

6. Figures 5 and 6. Consider inverting the order of the figures, and describe them more precisely in the text (e.g., "Fig. 5 show retrieved size distributions and concentration profiles for the case study etc...., and Fig. 6 shows the agreement betwen observed values of total optical depth, sky radiance and lidar signals, and the values fitted with GARRLic/GRASP).

   **Response:** We prefer not to change the order of the figures since we think that it follows better the order of the retrieval process, i.e. first the fitting of the measurements and then the provision of the results. Following the reviewer's suggestion we describe more the figures in the text, changing the lines 298-299 as following: **"*Figure 5 shows an example of the fitting with GRASP/GARRLiC of the lidar and sun-photometer observations (i.e., the total optical depth, sky radiance and lidar signals) for one of the acceptable solutions (size distributions and concentration profiles), which is shown in Fig. 6.*"**

7. Does Fig. 7 add any new information to Fig. 6? If not, you may consider removing it (isn't the optimal solution already highlighted in Fig. 6?)

   **Response:** Thank you for the suggestion. We have removed Fig. 7.

8. L306-307. "This indicates... part of the retrieval uncertainty of the solution". Is this a general statement or does it just hold for the example you are showing?

   **Response:** Thank you for the comment. It is a general statement for the case studies investigated within our work. We added the following in line 307 to make this clearer: *"This conclusion holds for all the case studies investigated herein."*

9. L330. "the information content... size distribution" -> "the information content of lidar measurements on the aerosol size distribution"

   **Response:** Thank you, it has been corrected.

10. L339. Some references are missing here. Are you maybe referring to Fig. 10?

    **Response:** Thank you, it has been corrected.

11. L340, eq. 1-2, Appendix B. Why do you indicate concentration with VD? Could you use a more intuitive symbol? For example, in eq. 3 you use c...

    **Response:** Thank you, we have changed it to "VC".

12. L369-372. "h_a (pink)... full overlap height". What do you mean by "taking into account"? How do you actually take those values into account?

13. **Response:** Thank you, we have changed lines 369-372 as follows: *"…"ha" (pink) is the concentration profile comprised by the maximum values above the full overlap height and the minimum values below the full overlap height. "hb" (dark blue) is the concentration profile comprised by the minimum values above the full overlap height and the maximum values below the full overlap height."*

14. Figs. 14.-15. How do the variables that appear in the plot correlate? Scatter plots may be useful in addition to histograms.

    **Response:** We provide here the scatterplots of the differences (GRASP/GARRLiC – AERONET) for the effective radius vs the volume concentration (left), along with the corresponding normalized differences (right):

[Figure]

We see an anti-correlation between the differences in the retrieved effective radius and the differences in the retrieved volume concentration. Thus, the smaller effective radii retrieved by GRASP/GARRLiC is associated with higher values for the volume concentration. This anti-correlation is not easy to interpret. We included these scatter plots in the manuscript as Fig. 15 and we added the following in line 388: "*Figure 15 shows the scatterplots of the differences (GRASP/GARRLiC – AERONET) of the retrieved effective radius vs the retrieved volume concentration, along with the corresponding normalized differences. The plots show that the differences are anti-correlated, with the lower values of the effective radius retrieved from GRASP/GARRLiC to be associated to higher values of retrieved volume concentrations. This anti-correlation is not easy to interpret.*"

15. L426. "depends also" -> "also depends"

     **Response:** Thank you, it has been corrected.

16. L460. "of the simulated signals against the input ones" -> "between simulated and measured signals"

     **Response:** Thank you, it has been corrected.

17. L462, "to obtain a quantitative metric" -> "to obtain such a metric", "input" -> "measured"?

     **Response:** Thank you, it has been corrected.

---

## Referee Report (RR1)

Minor comments:

1. Line 309:  Looking at Figure 3b, the fine mode fraction is higher than 75% rather than the 80% in the paper.
2. Line 358: What is the retrieval uncertainty of the AERONET size distribution, and does the optimum retrieval fall within this uncertainty?
3. Line 382: Likewise, what are the retrieval uncertainties for the AERONET results shown in Figure 10?
4. Line 384: What results are not statistically significant? Do you mean the differences in the size distribution between stations are not statistically significant?
5. Line 390: The symbols in Figure 12a are hard to distinguish.

---

## Author Response (AR2)

We would like to thank you a lot for your time to evaluate our manuscript and for your valuable feedback. Below you may find our replies to your comments written with blue colour. New or changed sentences in our manuscript are also provided (in black colour/italics).

**Minor comments**

1. Line 309: Looking at Figure 3b, the fine mode fraction is higher than 75% rather than the 80% in the paper

   **REPLY:** Thank you, it has been corrected.

   **Lines 298-300:** "*According to the collocated sunphotometer measurements, the AOD at 500 nm does not exceed the value of 0.15 and the fine aerosol fraction is higher than 75% (Fig. 3b).*"

2. Line 358: What is the retrieval uncertainty of the AERONET size distribution, and does the optimum retrieval fall within this uncertainty?

   **REPLY:** As described in Dubovik et al. (2000), the relative retrieval error of the volume size distribution of water soluble particles, with radius in the range of 0.1-7 μm, is 15%. The differences between GRASP/GARRLiC and AERONET retrievals are larger than this threshold for some of the cases analyzed. We inserted the following in line 373: "*The differences between GRASP/GARRLiC and AERONET retrievals are larger than 15% (i.e., the retrieval uncertainty of the AERONET product provided by Dubovik et al. (2000) for water-soluble particles with radius in the range of 0.1-7 μm), for a number of cases analyzed.*"

   **Reference**

   Dubovik, O., Smirnov, A., Holben, B., King, M., Kaufman, Y., Eck, T., and Slutsker, I.: Accuracy assessments of aerosol optical properties retrieved from Aerosol Robotic Network (AERONET) Sun and sky radiance measurements, J. Geophys. Res.-Atmos., 105, 9791–9806, https://doi.org/10.1029/2000jd900040, 2000.

3. Line 382: Likewise, what are the retrieval uncertainties for the AERONET results shown in Figure 10?

   **REPLY:** See reply of previous comment.

4. Line 384: What results are not statistically significant? Do you mean the differences in the size distribution between stations are not statistically significant??

**REPLY:** We mean that the number of cases presented are not sufficient for characterizing the atmospheric state above Europe during the COVID-19 lockdown and relaxation period.

We changed lines 373-374 accordingly: "*Due to the low number of cases, the results are not statistically significant in order to characterize the atmospheric state over Europe during the COVID-19 lockdown and relaxation period.*"

5. Line 390: The symbols in Figure 12a are hard to distinguish.

**REPLY:** Thank you for the suggestion. We updated Fig. 12a, using the following plots: